# IBIA: An Incremental Build-Infer-Approximate Framework for Approximate Inference of Partition Function

**Shivani Bathla**                                                                  *ee13s064@ee.iitm.ac.in*
*Department of Electrical Engineering*
*IIT Madras*

**Vinita Vasudevan**                                                                *vinita@ee.iitm.ac.in*
*Department of Electrical Engineering*
*IIT Madras*

**Reviewed on OpenReview:** *https://openreview.net/forum?id=8L7Rh6FIXt*

## Abstract

Exact computation of the partition function is known to be intractable, necessitating approximate inference techniques. Existing methods for approximate inference are slow to converge for many benchmarks. The control of accuracy-complexity trade-off is also non-trivial in many of these methods. We propose a novel *incremental build-infer-approximate* (IBIA) framework for approximate inference that addresses these issues. In this framework, the probabilistic graphical model is converted into a sequence of clique tree forests (SCTF) with bounded clique sizes. We show that the SCTF can be used to efficiently compute the partition function. We propose two new algorithms which are used to construct the SCTF and prove the correctness of both. The first is an algorithm for incremental construction of CTFs that is guaranteed to give a valid CTF with bounded clique sizes and the second is an approximation algorithm that takes a calibrated CTF as input and yields a valid and calibrated CTF with reduced clique sizes as the output. We have evaluated our method using several benchmark sets from recent UAI competitions and our results show good accuracies with competitive runtimes.

## 1 Introduction

Discrete probabilistic graphical models including Bayesian networks (BN) and Markov networks (MN) have been used for probabilistic inference in a wide variety of applications. A fundamental task in inference is the computation of the partition function (PR), which is the normalization constant for the overall probability distribution. Exact inference of PR is known to be #P complete (Roth, 1996), necessitating approximations in general. Methods for approximate inference can be broadly classified as methods based on variational optimization and sampling or search based methods.

Variational techniques cast the inference problem as an optimization problem, which is typically solved using iterative message-passing algorithms. These include loopy belief propagation (LBP) (Murphy et al., 1999; Wainwright et al., 2002; Wiegerinck & Heskes, 2003), region-graph based methods like generalized belief propagation (GBP) and its variants (Yedidia et al., 2000; Heskes, 2006; Mooij & Kappen, 2007; Sontag et al., 2008; Lin et al., 2020), mean-field approximations (Winn et al., 2005) and methods based on expectation propagation (Minka, 2001; 2004; Vehtari et al., 2020). A combination of mini-bucket heuristics and belief propagation is used in methods like weighted mini-bucket elimination (WMB) (Liu & Ihler, 2011; Forouzan & Ihler, 2015; Lee et al., 2020) and iterative join graph propagation (IJGP) (Mateescu et al., 2010). While the parameter settings for complexity accuracy trade-off is non-trivial in many of the GBP based methods, it is controlled using a single user-defined parameter (*ibound*) in mini-bucket based methods. More recently, several extensions of mini-bucket based methods have been proposed. These include bucket re-normalization

(Ahn et al., 2018), deep-bucket elimination (DBE) (Razeghi et al., 2021) and NeuroBE (Agarwal et al., 2022). Both DBE and NeuroBE use neural networks to improve the quality of approximations. Another approach is to bound clique sizes by simplifying the network. In the thin junction tree based methods (Bach & Jordan, 2001; Elidan & Gould, 2008; Scanagatta et al., 2018), a set of features (nodes and edges) so that the resulting graph has a bounded tree-width. The remaining features are ignored. The edge deletion belief propagation (EDBP) and the related relax-recover compensate (RRC) methods (Choi et al., 2005; Choi & Darwiche, 2006; 2007; 2008) perform inference on progressively more complex graphs in which new features are added, while satisfying some consistency conditions.

Sampling algorithms can be classified as methods based on Markov chain Monte Carlo (MCMC) like Gibbs sampling (Gelfand, 2000) and methods based on importance and stratified sampling (Bouckaert et al., 1996; Hernandez et al., 1998; Moral & Salmerón, 2005). The more recent importance sampling based methods use proposals based on approximate variational methods like WMB and IJGP. In Liu et al. (2015), WMB is used as the proposal for importance sampling (WMB-IS). The dynamic importance sampling (DIS) method proposed in Lou et al. (2017) also uses WMB and has a periodic update of the sampling proposal. The abstraction sampling methods (Broka, 2018; Kask et al., 2020) use an abstraction function to merge similar nodes in AND-OR search trees to get abstract states. An estimate of the PR is obtained using sampled subtrees, with WMB used in the sampling proposal. Sample search (Gogate & Dechter, 2007) is a variant of importance sampling that deals with the rejection of samples in the presence of zero weights. The method proposed in Gogate & Dechter (2011) uses sample search with cutset sampling and an IJGP based proposal. Another approach is to combine sampling techniques with model counting based methods (Chakraborty et al., 2013; 2016; Soos & Meel, 2019; Sharma et al., 2019).

**Limitations of existing methods:** Sampling based methods are anytime algorithms where it is possible to improve accuracy by increasing the number of samples, without the associated increase in memory. However, the performance of these methods depends significantly on the proposal distribution used for importance sampling. The results in Gogate & Dechter (2011); Lou et al. (2017; 2019); Kask et al. (2020) also indicate that after an initial rapid increase, the improvement in accuracy slows down significantly with time. Variational techniques typically require increase in both time and memory for better accuracy. LBP works with minimal cluster sizes and is therefore fast and gives solutions for most benchmarks (Agrawal et al., 2021). However, it results in poor accuracies especially for many of the harder benchmarks. The accuracy of GBP based methods depends on the choice of the outer regions, which is non-trivial. In practice, we have found that these methods are slow to converge for many benchmarks. Methods based on mini-bucket heuristics like WMB, WMB-IS and DIS have easy control of accuracy complexity trade-off but the accuracy obtained is often limited (Broka, 2018; Kask et al., 2020; Agarwal et al., 2022). Neural network based extensions like NeuroBE and DBE improve the accuracy of estimates, but require several hours of training. Selection of optimum features in the RRC and related methods is compute-intensive since it is based on metrics that require several iterations of belief propagation. While weighted model counting based methods work well for many benchmarks, they struggle for benchmarks with large variable domain cardinality (Agrawal et al., 2021).

**Contributions of this paper**: In this paper, we propose a new framework for approximate inference that addresses some of these issues. Our framework, denoted the *incremental build-infer-approximate* (IBIA) paradigm, converts each connected graph in the PGM into a data structure that we call *Sequence of Clique Tree Forests* (SCTF). We show that the SCTF can be used for efficient computation of the PR. To construct the SCTF, we propose two new algorithms and prove the correctness of both. The first is an algorithm for incremental construction of CTFs that is guaranteed to give a valid CTF with bounded clique sizes and the second is an approximation algorithm that takes a calibrated CTF as input and yields a valid and calibrated CTF with reduced clique sizes as the output.

Our method has easy control of accuracy-complexity trade-off using two user-defined parameters for clique size bounds, which are similar to the *ibound* parameter setting in mini-bucket based methods. Since IBIA is based on clique trees and not loopy graphs, the belief propagation step is non-iterative and there are no convergence issues. In IBIA, approximations are based on clique beliefs and not the network structure alone, which results in good accuracies. We evaluated our method with 1717 instances belonging to different benchmark sets included in several UAI competitions. Results show that the accuracy of solutions obtained

by IBIA is better than the other variational techniques. It also gives comparable or better accuracies than the state of art sampling methods in a much shorter time.

**Organization of this paper:** The rest of this paper is organized as follows. Section 2 provides background and notation. We present an overview of the IBIA framework in Section 3, the methodology for constructing the SCTF in Section 4 and approximate inference of PR in Section 5. We present the complexity analysis in Section 6, results in Section 7 and comparison with related work in Section 8. Finally, we present our conclusions in Section 9. The proofs for all propositions and theorems are included in Appendix A.

## 2 Background

This section has the notation and the definitions used in this paper.

**Definition 1. Probabilistic graphical model (PGM):** Let $\mathcal{X} = \{X_1, X_2, \cdots X_n\}$ be a set of random variables with associated domains $D = \{D_{X_1}, D_{X_2}, \cdots D_{X_n}\}$. The probabilistic graphical model (PGM) over $\mathcal{X}$ consists of a set of factors, $\Phi$. Each factor $\phi_\alpha(\mathcal{X}_\alpha) \in \Phi$ is defined over a subset of variables $Scope(\phi_\alpha) = \mathcal{X}_\alpha$, where $\alpha$ denotes the index to the set of factors. The domain $D_\alpha$ of $\mathcal{X}_\alpha$ is the Cartesian product of the domains of variables in $\mathcal{X}_\alpha$ and the factor $\phi_\alpha$ is a map $\phi_\alpha : D_\alpha \to R \geq 0$. The joint probability distribution captured by the model is $P(\mathcal{X}) = \frac{1}{Z} \prod_\alpha \phi_\alpha$ where the normalizing constant, $Z = \sum_{Domain(\mathcal{X})} \prod_\alpha \phi_\alpha$ is the partition function (PR).

Each node of the undirected graph corresponding to the PGM is associated with a random variable, $X_i \in \mathcal{X}$. Variables $X_i$ and $X_j$ are connected via an edge in this graph if there is at least one factor in the PGM ($\Phi$) whose scope contains both variables.

**Definition 2. Chordal graph ($\mathcal{H}$):** It is an undirected graph with no cycle of length greater than three.

**Definition 3. Clique:** A subset of nodes in an undirected graph such that all pairs of nodes are adjacent.

**Definition 4. Maximal clique:** A clique that is not contained within any other clique in the graph.

**Definition 5. Junction tree or Join tree or Clique tree (CT)** (Koller & Friedman, 2009): The CT is a hypertree with nodes that are cliques ($C_i$) in the chordal graph ($\mathcal{H}$) corresponding to the undirected graph of the PGM. An edge between $C_i$ and $C_j$ is associated with a sepset $S_{i,j} = C_i \cap C_j$. A valid CT satisfies the following properties.

(a) All cliques are maximal cliques i.e., there is no $C_j$ such that $C_j \subset C_i$.

(b) It satisfies the running intersection property (RIP), which states that for all variables $X$, if $X \in C_i$ and $X \in C_j$, then $X$ is present in every node in the unique path between $C_i$ and $C_j$.

(c) Each factor $\phi_\alpha$ in the PGM is assigned to a single node $C_i$ such that $Scope(\phi_\alpha) \subseteq C_i$.

Note that throughout the paper, we use the terms clique tree, join tree and junction tree interchangeably. Also, as is common in the literature, we use the term $C_i$ as a label for the node in the CT as well as to denote the set of variables in the clique.

The initial belief associated with clique $C_i$ is the product of all factors assigned to $C_i$. Exact inference in a CT is done using the belief propagation (BP) algorithm (Lauritzen & Spiegelhalter, 1988; Koller & Friedman, 2009) that is equivalent to two rounds of message passing along the edges of the CT, an upward pass (from the leaf nodes to the root node) and a downward pass (from the root node to the leaves). Following this, each clique in the CT has an associated joint belief $\beta(C_i) = \sum_{Domain(\mathcal{X}\setminus C_i)} \prod_\alpha \phi_\alpha$ and each sepset has an associated joint belief $\mu(S_{ij}) = \sum_{Domain(\mathcal{X}\setminus S_{ij})} \prod_\alpha \phi_\alpha$.

**Definition 6. Calibrated CT** (Koller & Friedman, 2009): Let $\beta(C_i)$ and $\beta(C_j)$ denote the beliefs associated with adjacent cliques $C_i$ and $C_j$. The cliques are said to be calibrated if

$$\sum_{Domain(C_i\setminus S_{i,j})} \beta(C_i) = \sum_{Domain(C_j\setminus S_{i,j})} \beta(C_j) = \mu(S_{i,j}) \tag{1}$$

Here, $S_{i,j}$ is the sepset corresponding to $C_i$ and $C_j$, and $\mu(S_{i,j})$ is the associated sepset belief. The CT is said to be calibrated if all pairs of adjacent cliques are calibrated. It has the following properties.

(a) All clique and sepset beliefs in the calibrated CT have the same normalization constant ($Z$) which is equal to the partition function (PR).

(b) The joint probability distribution, $P(\mathcal{X})$, can be re-parameterized in terms of the sepset and clique beliefs as follows:

$$P(\mathcal{X}) = \frac{1}{Z} \frac{\prod_{i \in \mathcal{V}_T} \beta(C_i)}{\prod_{(i,j) \in \mathcal{E}_T} \mu(S_{i,j})} \tag{2}$$

where $\mathcal{V}_T$ and $\mathcal{E}_T$ are the set of nodes and edges in the CT.

## 3 Overview of the IBIA paradigm

This section has the definitions of terms used in various algorithms and an overview of the IBIA paradigm. We also introduce a running example that will be used in various sections of this paper to illustrate the constituent algorithms.

### 3.1 Definitions

We use the following definitions in the paper.

**Definition 7. Clique Tree Forest (CTF):** Set of disjoint CTs.

**Definition 8. Valid CTF:** A CTF is valid if all CTs in the CTF are valid i.e., they satisfy all properties in Definition 5.

**Definition 9. Calibrated CTF:** A CTF is calibrated if all CTs in the CTF are valid and calibrated.

**Definition 10. Clique size:** The clique size $cs_i$ of a clique $C_i$ is defined as follows.

$$cs_i = \log_2 \left( \prod_{v \in C_i} |D_v| \right) \tag{3}$$

where $|D_v|$ is the cardinality or the number of states in the domain of the variable $v$.

It can be seen from the definition that the clique size is the effective number of binary variables contained in the clique.

### 3.2 Motivation

Since the complexity of inference is exponential in the maximum clique size, the key to making the problem tractable is to bound the clique size. Typically, bounding clique sizes leads to loopy graphs and convergence issues. An alternative is to divide the PGM into multiple sections such that each section results in a CTF with smaller clique sizes, thus making it amenable to non-iterative belief propagation. Existing approaches are multiply sectioned Bayesian networks (MSBN) (Xiang et al., 1993; Xiang & Lesser, 2003), which is an exact inference method and the approximate inference method proposed in Bhanja & Ranganathan (2004). The limitations of these methods are discussed in Section 8. Following are the two main challenges that need to be addressed: (a) How do we divide the PGM such that the maximum clique size of each CTF is less than a user-specified bound? (b) How do we exchange beliefs between the CTFs, so that the overall partition function can be inferred?

These two challenges are addressed in this paper.

### 3.3 Overview

The inputs to the algorithm are the set of initial factors ($\Phi$) and two user-defined clique size parameters $mcs_p$ and $mcs_{im}$. Let $\mathcal{G}$ denote the undirected graph induced by $\Phi$. Figure 1 illustrates the overall methodology used in IBIA to construct the sequence of CTFs (SCTF) for $\mathcal{G}$ and get an estimate of the partition function for the given set of factors. The three main steps in the method are as follows.
**Incremental Build**: Starting with a valid CTF, the algorithm (Algorithm 1) builds the CTs in the CTF

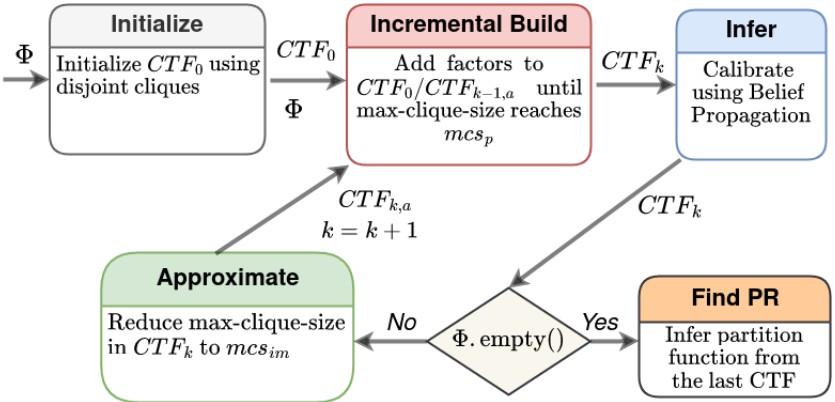

Figure 1: Estimation of partition function using the IBIA framework

by incrementally adding new factors to it as long as the maximum clique size bound, $mcs_p$, is not violated. We show that the result of Algorithm 1 is guaranteed to be a valid CTF. It is assumed that $mcs_p$ is large enough to accommodate the maximum domain size of the factors.

**Infer**: This step takes a valid CTF as input and calibrates all the CTs in the CTF using the standard BP algorithm (Lauritzen & Spiegelhalter, 1988) for exact inference. After calibration, all clique beliefs in a CT have the same normalization constant.

**Approximate**: The input to this algorithm (Algorithm 2) is a calibrated CTF, $CTF_k$, with maximum clique size $mcs_p$. The result of the algorithm is an approximate CTF, $CTF_{k,a}$, with a reduced maximum clique size of $mcs_{im}$. Our approximation algorithm ensures that $CTF_{k,a}$ is valid and calibrated so that the CTs need not be re-calibrated using the message-passing algorithm. It also ensures that a connected CT in $CTF_k$ remains connected in $CTF_{k,a}$ and normalization constants of the CTs in the CTF are unchanged.

Assume that $\mathcal{G}$ is connected. The construction of the SCTF starts with an initial CTF ($CTF_0$) that contains cliques corresponding to factors in $\Phi$ with disjoint scopes. As shown in the figure, the three steps incremental build, infer and approximate are used repeatedly to construct the SCTF = $\{CTF_1, \cdots, CTF_n\}$. The construction is complete once all factors in $\Phi$ have been added to some CTF in the SCTF. The SCTF is thus a sequence of calibrated CTFs, each of which satisfies a property proved in Proposition 9. Based on this property, we show that the last CTF, $CTF_n$, contains a single connected CT and the normalization constant of this CT is the estimate of the PR (Theorem 2).

If $\mathcal{G}$ has multiple disjoint graphs, which happens for example after evidence based simplification, an SCTF is constructed for each connected graph and the estimate of PR is the product of the normalization constants of the last CTF of each SCTF.

### 3.4 Example

We will use the example shown in Figure 2a as a running example to explain the steps used in various algorithms proposed in this work. The figure has the factors and the input graph induced by the factors. All variables are assumed to be binary and $mcs_p$ and $mcs_{im}$ are set to 4 and 3 respectively. The final result is an SCTF consisting of two CTFs shown in Figure 2b. The normalization constant of clique beliefs in $CTF_2$ is the estimated PR for the example.

## 4 Construction of the SCTF

In this section, we describe the three steps that are used to generate the sequence of CTFs namely, incremental build, infer and approximate. We use the following definitions in this section.

**Definition 11. $MSG[V]$:** Given a subset of variables ($V$) in a valid CTF, $MSG[V]$ is used to denote the minimal subgraph of the CTF that is needed to compute the joint beliefs of $V$.

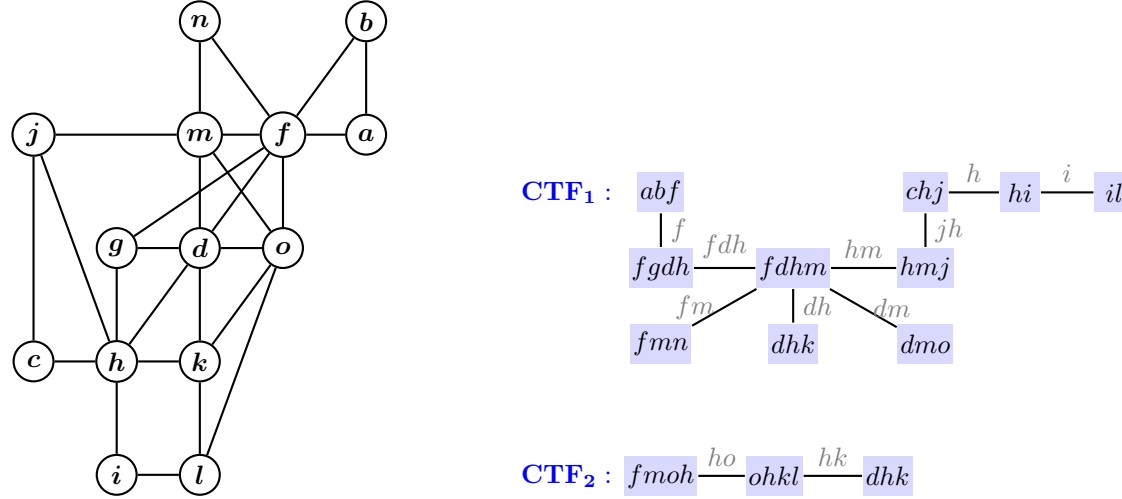

(a) Undirected graph induced by input set of factors Φ.        (b) Corresponding sequence of CTFs (SCTF).

Figure 2: Construction of sequence of CTFs (SCTF) for the input set of factors $\Phi = \{\phi(c, h, j),\ \phi(f, g, d),\ \phi(i, l), \phi(a, b, f),\ \phi(d, g, h),\ \phi(d, h, k),\ \phi(h, i),\ \phi(f, o),\ \phi(j, m), \phi(f, m, n),\ \phi(d, m, o),\ \phi(k, l, o)\}$. The maximum clique size constraints, $mcs_p$ and $mcs_{im}$ are set to 4 and 3 respectively. All variables are assumed to be binary.

It is obtained by first identifying the subgraph of CTF that connects all the cliques that contain variables in the set $V$. Then, starting from the leaf nodes of the subgraph (nodes with degree equal to 1), cliques that contain the same set (or subset) of variables in $V$ as their neighbors are removed recursively.

**Definition 12. Interface variables (IV):** Let the initial set of factors in the PGM be $\Phi = \{\Phi_1, \ldots, \Phi_n\}$, where $\Phi_k$ denotes the set of factors added to $CTF_k$. A variable in $CTF_k$ is an interface variable if it is present in the scope of any factor in the set $\{\Phi_{k+1}, \ldots, \Phi_n\}$.

Each CTF in the sequence has a different set of interface variables. IVs are needed to form the next CTF in the sequence.

## 4.1 Incremental Build

In this step, new factors from a set $\Phi$ are incrementally added to an existing valid CTF, which is either $CTF_0$ or the approximate CTF, $CTF_{k-1,a}$, as long as the maximum clique size bound ($mcs_p$) is not violated. If the scope of a new factor is a subset of an existing clique, the factor is simply assigned to the clique. Otherwise, we need to modify the CTF to add a clique that contains the scope of the new factor while ensuring that the CTF remains valid. We first explain our method of construction of CTFs with the help of the running example. We then formally state the steps and prove the correctness of our algorithm.

### 4.1.1 Example

Figure 3 illustrates the construction of $CTF_1$ from an initial CTF, $CTF_0$. $CTF_0$ is initialized with cliques corresponding to factors with disjoint scopes, chosen as cliques $C_1, C_3$ and $C_9$ in the example. These are highlighted in red in the graph. Let $\mathcal{V}$ denote the set of variables present in the existing CTF. The first step in the addition of a factor $\phi$ is the identification of the subgraph $SG_{min} = MSG[scope(\phi) \cap \mathcal{V}]$. The method for addition of $\phi$ to the CTF depends on whether $SG_{min}$ is a set of disjoint cliques or it has connected components. The steps involved in the two cases are as follows.

**1. $SG_{min}$ is a set of disjoint cliques**: Assume that the factor $\phi(h, i)$ is to be added to $CTF_0$. In this case, $SG_{min} = MSG[h, i]$ consists of two disjoint cliques, $C_3$ and $C_9$. As shown in the figure, the new clique corresponding to $\phi(h, i)$ can simply be connected to cliques $C_3$ and $C_9$ via the sepset variables $h$ and $i$, producing a valid CTF. The addition of factor $\phi(d, g, h)$ is similar. $SG_{min}$ for the factors $\phi(d, h, k), \phi(a, b, f)$

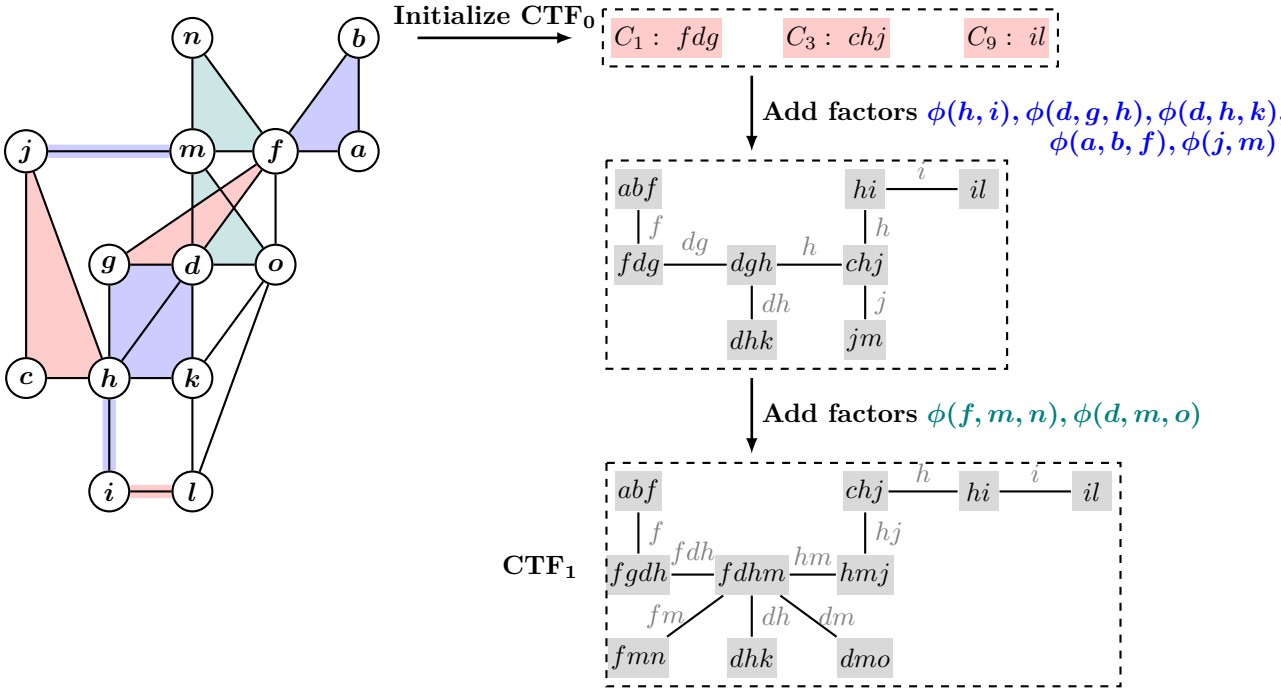

Figure 3: Construction of the first CTF in the sequence, $CTF_1$, for an example PGM with $mcs_p$ set to 4. Starting with a set of disjoint cliques, factors are added until the maximum clique size reaches $mcs_p$. Factors $\phi(k, l, o)$ and $\phi(f, o)$ are deferred for addition to the next CTF.

and $\phi(j, m)$ are single cliques. As shown in the figure, they can be connected to the existing CTF via the corresponding sepsets to produce a valid CTF.

**2. $SG_{min}$ has connected components**: When we try to add factor $\phi(f, m, n)$, the variables $f$ and $m$ are present in cliques $C_5$ and $C_4$ which are already connected in the existing CTF. Directly connecting these cliques to the new clique containing variables $f$, $m$ and $n$ will generate a loop and hence result in an invalid CTF. Figure 4 shows the steps used for addition of this factor. $SG_{min} = MSG[\{f, m\}]$ is highlighted in red in Figure 4a. The goal is to replace $SG_{min}$ with a subtree $ST'$ that has a clique containing variables $f$, $m$ and $n$, while ensuring that the resulting CTF remains valid. As shown in Figure 4b, when the new clique is added to the chordal graph corresponding to $SG_{min}$, chordless loops $f$-$g$-$h$-$j$-$m$-$f$ and $f$-$d$-$h$-$j$-$m$-$f$ are introduced. Therefore, retriangulation is needed to get back a chordal graph. However, only a subgraph of the modified chordal graph needs to be re-triangulated. Using variable elimination to form cliques, clique containing variables $c$, $h$ and $j$ is obtained after eliminating variable $c$. This clique is already present in $SG_{min}$. We call such cliques as *retained cliques*. The subgraph $G_E$ shown in Figure 4c is obtained after removing the variable $c$ and deleting the corresponding edges. This is the subgraph that needs re-triangulation. We call it the *elimination graph* and denote the variables in this graph as the *elimination set* ($S_E$). Comparing Figures 4a and 4c, we see that $S_E$ contains the sepset variables in $SG_{min}$ and the variables in the new factor. On triangulating $G_E$, we get a CT, $ST'$, that contains cliques $C_1'$, $C_2'$, $C_3'$ and $C_4'$ as shown in Figure 4d. Each retained clique is then connected to a clique in $ST'$ such that the sepset contains all common variables. In the example, clique $C_3$ gets connected to clique $C_3'$ via sepset variables $h$ and $j$ which are present in both $C_3$ and $ST'$. The final $ST'$ is highlighted in teal in Figure 4d. $ST'$ replaces $SG_{min}$ in the existing CTF. The connection is done via cliques $C_5, C_7$ and $C_8$ that were adjacent to $SG_{min}$ with the same sepsets. Since cliques $C_1, C_2, C_4$ are no longer present in the modified CT, the associated factors are re-assigned to corresponding containing cliques in $ST'$. Accordingly, the factors associated with $C_1$ and $C_2$ are re-assigned to $C_1'$ and that associated with $C_4$ is re-assigned to $C_3'$. The new factor $\phi(f, m, n)$ is assigned to clique $C_4'$ that contains all variables in the scope of this factor.

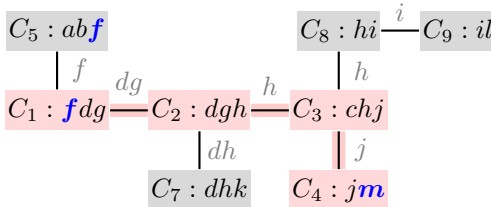

(a) Existing CTF. The minimal subgraph corresponding to variables in the new factor, $SG_{min}$, is highlighted in red.

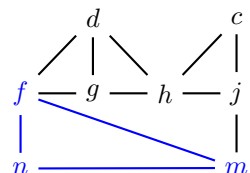

(b) Addition of a clique between $f, m, n$ to the chordal graph corresponding to $SG_{min}$.

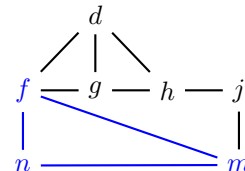

(c) The elimination graph, $G_E$, which is the subgraph of the modified chordal graph that needs re-triangulation.

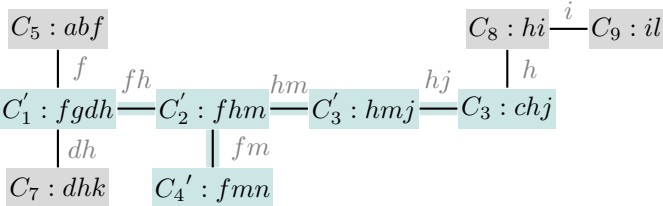

(d) The modified CTF obtained after replacing $SG_{min}$ with $ST'$ (marked in teal). $ST'$ contains cliques obtained after triangulating the elimination graph $(C_1', C_2', C_3', C_4')$ and the retained clique $C_3$.

Figure 4: Addition of a factor $\phi(f, m, n)$ to an existing CTF.

Factor $\phi(d, m, o)$ is added in a similar manner and the resulting CTF, $CTF_1$, is shown in Figure 3. Addition of factors $\phi(f, o)$ and $\phi(k, l, o)$ violates the clique size bounds ($mcs_p = 4$) and are deferred for addition to the next CTF in the sequence. Note that $ST'$ is not unique and depends on the elimination order used for re-triangulation. Similarly, the replacement of $SG_{min}$ by $ST'$ can be done in multiple ways. Therefore, the resulting CTF is not unique, but it is always a valid CTF.

Often the new factors that need to be added impact overlapping portions of the existing CTF. While they can be added sequentially, adding them together not only reduces the effort required for re-triangulation, but also often results in smaller clique sizes. Therefore, in our algorithm factors having overlapping $SG_{min}$ are added together as a group. The procedure to add a group of factors is similar.

### 4.1.2 Algorithm

We first define various terms used in the algorithm. Let $\mathcal{V}$ denote the variables in the existing CTF, $\Phi$ denote the set of factors to be added and $Scope(\Phi) = \cup_{\phi \in \Phi} Scope(\phi)$.

**Definition 13. $SG_{min}$:** It is defined as $MSG[Scope(\Phi) \cap \mathcal{V}]$ (see Definition 11 for $MSG$).

It is the minimal portion of the existing CTF that is impacted by the addition of new factors.

**Definition 14. Elimination set ($S_E$):** It is the set containing the variables in the new factors to be added and the variables in the sepsets of $SG_{min}$.

**Definition 15. Retained cliques:** Cliques in $SG_{min}$ that contain variables that are not contained in the set $S_E$.

**Definition 16. Elimination graph ($G_E$):** The elimination graph is constructed using the following steps:-
(a) For each factor $\phi$ in the set $\Phi$, add a fully connected component between variables in $Scope(\phi)$ (b) For each clique $C \in SG_{min}$, add a fully connected component corresponding to $C \cap S_E$.

Algorithm 1 shows the formal steps in our algorithm for incremental addition of new factors to an existing CTF such that clique sizes are bounded. The inputs to the algorithm are a valid CTF, the set of factors to be added ($\Phi$) and the clique size bound $mcs_p$. In each step of this algorithm, we attempt to add a group of factors that have overlapping $SG_{min}$ ($\Phi_g$) (lines 3-15). To do this, we first find the $SG_{min}$ corresponding

to the entire group $\Phi_g$ (Definition 13) and construct the modified subtree $ST'$ by adding factors in $\Phi_g$ to $SG_{min}$ (lines 6-7). If $ST'$ satisfies the clique size bounds, the CTF is modified and the $\Phi_g$ is removed from $\Phi$ (lines 9-11). Otherwise, we remove a subset of factors, $\Phi_{gs}$, from $\Phi$ and try adding the remaining factors to the CTF. $\Phi_{gs}$ is added to $\Phi_d$, which is a list of factors that are deferred for addition to subsequent CTFs

---

**Algorithm 1** BuildCTF($CTF, \Phi, mcs_p$)

---

**Input:** $CTF$: Input CTF
$\Phi$: Set of new factors to be added
$mcs_p$: Maximum clique size bound for the modified CTF
**Output:** $CTF$: Modified CTF
$\Phi$: Set of remaining factors
1: **Initialize:** $\Phi_d = \{\}$ ▷ Set of factors deferred for addition to subsequent CTFs
2: **while** $\Phi.isNotEmpty()$ **do** ▷ Loop until further addition is not possible
3:     $\mathcal{V} = \{Variables \in CTF\}$
4:     For each factor $\phi \in \Phi$, identify the corresponding minimal subgraph $SG_{min} = MSG[Scope(\phi) \cap \mathcal{V}]$
5:     $\Phi_g \leftarrow$ Find a group of factors with overlapping minimal subgraphs
6:     $SG_{min} \leftarrow MSG[Scope(\Phi_g) \cap \mathcal{V}]$ ▷ Find the minimal subgraph corresponding to set $\Phi_g$
7:     $ST' \leftarrow$ Construct $ST'$ ($\Phi_g$, $SG_{min}$) ▷ $ST'$: Modified subtree
8:     ▷ Modify $CTF$ if clique size bound is satisfied.
9:     **if** Max-clique-size($ST'$) $\leq mcs_p$ **then**
10:         $CTF \leftarrow$ Modify CTF($ST', SG_{min}, CTF$) ▷ Replace $SG_{min}$ with $ST'$ and get modified CTF
11:         $\Phi \leftarrow \Phi \setminus \Phi_g$ ▷ Update the set of remaining factors
12:     **else**
13:         $\Phi_{gs} \leftarrow \{$Subset of factors $\in \Phi_g\}$ ▷ Choose a subset of factors for addition to subsequent CTFs
14:         $\Phi \leftarrow \Phi \setminus \Phi_{gs}$; $\Phi_d.add(\Phi_{gs})$; ▷ Remove $\Phi_{gs}$ from $\Phi$ and add it to the set of deferred factors $\Phi_d$
15:     **end if**
16: **end while**
17: $\Phi = \Phi_d$
18:
19: **procedure** CONSTRUCT $ST'(\Phi_g, SG_{min})$
20:     Construct the elimination set $S_E$ and elimination graph $G_E$, as per Definitions 14 and 16
21:     $ST' \leftarrow$ Triangulate $G_E$ and find the corresponding clique tree
22:     ▷ Identify the set of retained cliques, $\mathcal{C}_r$
23:     $\mathcal{V}_{sg} \leftarrow \{$Variables $\in SG_{min}\}$; $\mathcal{V}_r \leftarrow \mathcal{V}_{sg} \setminus S_E$ ▷ $\mathcal{V}_r$: Variables used to identify retained cliques
24:     $\mathcal{C}_r \leftarrow$ Cliques $\in SG_{min}$ that contain at least one variable in $\mathcal{V}_r$ ▷ $\mathcal{C}_r$: Set of retained cliques
25:     ▷ Connect retained cliques to $ST'$
26:     **for** $C \in \mathcal{C}_r$ **do**
27:         Find a clique $C' \in ST'$ such that $C \cap S_E \subseteq C'$
28:         **if** $C' \subset C$ **then** Replace $C'$ by $C$ **else** Connect $C'$ to $C$ ▷ Check maximality, connect retained clique $C$
29:     **end for**
30:     ▷ Assign factors to cliques in $ST'$
31:     Re-assign factors associated with cliques in $SG_{min}$ to containing cliques in $ST'$
32:     Assign factors in $\Phi_g$ to containing cliques in $ST'$
33:     **return** $ST'$
34: **end procedure**
35:
36: **procedure** MODIFY CTF($ST', SG_{min}, CTF$)
37:     ▷ Replace $SG_{min}$ with $ST'$ in CTF
38:     $Adj(SG_{min}) \leftarrow$ List of tuples $(C_a, S_a)$ containing cliques adjacent to $SG_{min}$ and corresponding sepset variables
39:     Remove $SG_{min}$ from CTF
40:     **for** $(C_a, S_a) \in Adj(SG_{min})$ **do** ▷ Re-connect cliques adjacent to $SG_{min}$ to cliques in $ST'$
41:         Connect $C_a$ to clique $C'$ in $ST'$ such that $S_a \subset C'$
42:     **end for**
43:     **return** $CTF$
44: **end procedure**

---

(lines 12-15). This process is continued until $\Phi$ becomes empty and no further addition is possible. After the CTF is built, we re-assign $\Phi$ to contain the set of all deferred factors (line 17).

*Construct ST' (lines 19-34)*: In this function, we first find the elimination set $S_E$ and the elimination graph $G_E$ as per Definitions 14 and 16. The elimination graph is then triangulated and the corresponding clique tree $ST'$ is obtained (lines 20-21). We then identify the set of retained cliques ($\mathcal{C}_r$) which contain variables that are not present in $S_E$ ($\mathcal{V}_r$) (lines 22-24). For each retained clique $C$, we find a clique $C'$ in $ST'$ that contains the set $C \cap S_E$. We show that this is always possible in the proof of Proposition 3. If $C'$ is a subset of $C$, we replace $C'$ with $C$. Otherwise, we connect $C$ to $C'$ (lines 25-29). Following this, factors associated with cliques in $SG_{min}$ are reassigned and new factors in $\Phi_g$ are assigned to corresponding containing cliques in $ST'$ (lines 30-33).

*Modify CTF (lines 36-44)*: This function modifies the CTF by replacing $SG_{min}$ by $ST'$. We start by finding the set of cliques adjacent to $SG_{min}$ in the input CTF ($Adj(SG_{min})$) and remove $SG_{min}$ from the CTF. Cliques in $Adj(SG_{min})$ are reconnected to cliques in $ST'$ that contain the corresponding sepset in the existing CTF. We show that this connection is always possible in the proof of Proposition 3.

### 4.1.3 Soundness of the algorithm

Let the input to Algorithm 1 be a valid CTF. Let $CTF_m$ denote the modified CTF obtained after adding a group of factors $\Phi_g$ to an existing CTF using lines 3-15 of Algorithm 1. Then the following propositions hold true. The proofs for these propositions are included in Appendix A.

**Proposition 1.** $CTF_m$ contains only trees (possibly disjoint) i.e., no loops are introduced by the algorithm.

**Proposition 2.** $CTF_m$ contains only maximal cliques.

**Proposition 3.** All CTs in $CTF_m$ satisfy the running intersection property (RIP).

**Proposition 4.** If the joint distribution captured by the input CTF with corresponding set of variables $X_{in}$ is $P(X_{in})$, then the joint distribution captured by $CTF_m$ is $P(X_{in}) \prod_{\phi \in \Phi_g} \phi$.

**Theorem 1.** Let the input CTF to Algorithm 1 be a valid CTF. Then, the CTF constructed by the algorithm is also a valid CTF with maximum clique size of $mcs_p$.

*Proof.* In Algorithm 1, we start with a valid CTF and sequentially add groups of factor using steps shown in lines 3-15. Based on Propositions 1 - 3, if the input is a valid CTF, the modified CTF is also a valid CTF since it satisfies all the properties needed to ensure that the CTF contains a set of valid CTs (see Definition 5). The clique size is bounded since the addition of factors is done only if the clique size bounds are met (line 9). □

### 4.2 Infer clique beliefs

The output of the incremental build step is a CTF, $CTF_k$, where the maximum clique size is at most $mcs_p$. In the *infer* step, $CTF_k$ is calibrated using the standard belief propagation algorithm for exact inference (Lauritzen & Spiegelhalter, 1988; Koller & Friedman, 2009). This is efficient since message passing is performed over clique trees with bounded clique sizes.

### 4.3 Approximate CTF

The next step is the *approximate* step, in which we reduce clique sizes in $CTF_k$ to get an approximate CTF, $CTF_{k,a}$. Based on Definition 12, we identify the interface variables (IV) in $CTF_k$. All the other variables in the CTF are referred to as *non-interface variables* (NIV). Since subsequent CTFs have factors that contain IVs, the accuracy of beliefs in these CTFs will depend on how well the joint beliefs of the IVs is preserved in $CTF_{k,a}$.

Figure 5 shows the steps required to get the approximate CTF ($CTF_{1,a}$) for the running example. In the example, $mcs_{im}$ is set to 3 and $IV = \{f, l, k, o\}$ (marked in red in the figure). $CTF_{1,a}$ is initialized to the minimal subgraph corresponding to IV, $MSG[\{f, l, k, o\}]$ (highlighted in blue in the figure). The two main

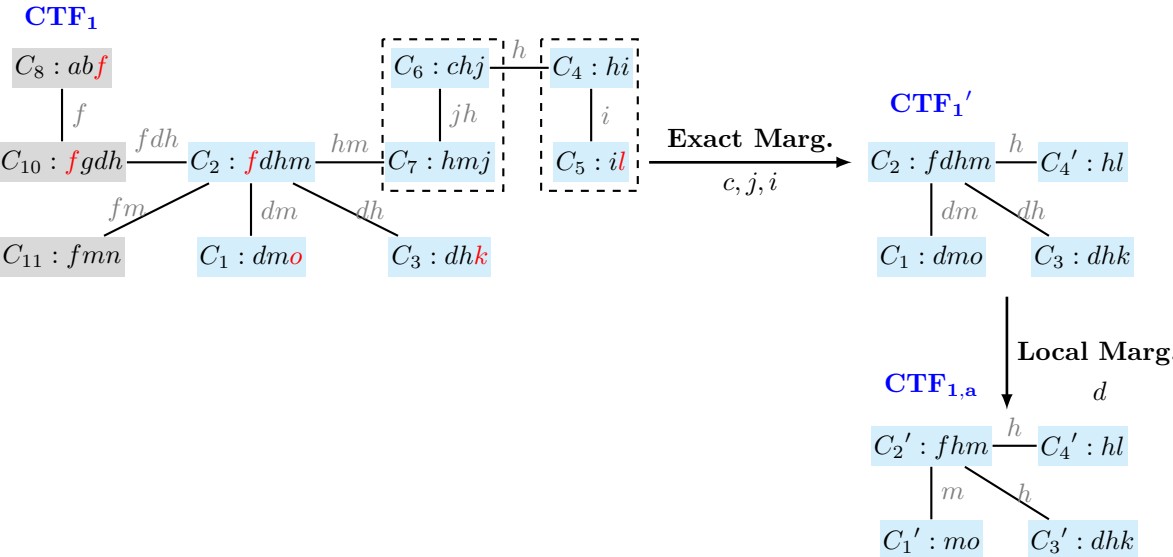

Figure 5: Approximation of $CTF_1$ for the running example with $mcs_{im}$ set to 3. The blue cliques in $CTF_1$ form the minimal subgraph corresponding to interface variables $f$, $k$, $o$ and $l$ (marked in red). $CTF_{1,a}$ is obtained after exact marginalization of non-interface variables $c, j, i$ and local marginalization of variable $d$.

steps used to reduce the clique sizes are exact and local marginalization, described below. For clarity, we explain the steps assuming that the clique sizes can be reduced exactly to the user-defined parameter $mcs_{im}$. In practice, it could be larger or smaller depending on the domain-sizes of the variables that are removed.

**Exact marginalization:** The goal of this step is to reduce the number of NIVs and the number of cliques in the CTF while preserving the joint beliefs over the IVs exactly. This can be done by removing some of the NIVs from the CTF as follows. If an NIV is present in a single clique, it is removed from the CTF and the corresponding clique belief is marginalized over all states in the domain of this variable. In case the resulting clique is non-maximal, it is removed and its neighbors are connected to the containing clique. If an NIV is present in multiple cliques, exact marginalization can only be done after collapsing all the cliques containing the variable into a single clique. Let $ST_v$ be the subtree of $CTF_{k,a}$ that has all the cliques containing a non-interface variable $v$ and $C_c$ be the new clique obtained after collapsing cliques in $ST_v$ and removing $v$. The clique belief for $C_c$ is obtained after marginalizing the joint probability distribution of $ST_v$ over all states in the domain of variable $v$, as follows.

$$\beta(C_c) = \sum_{D_v} \left( \frac{\prod_{C \in ST_v} \beta(C)}{\prod_{SP \in ST_v} \mu(SP)} \right) \tag{4}$$

where $SP$ denotes sepsets in $ST_v$ and $D_v$ denotes the domain of variable $v$. While this exactly preserves the joint distribution, this process becomes expensive or infeasible as the size of the collapsed clique increases. Therefore, we perform this step only if the size of the collapsed clique is less than or equal to $mcs_{im}$.

In the running example (shown in Figure 5), non-interface variable $c$ is present in a single clique $C_6$. It is removed from $C_6$ and the corresponding belief is marginalized. After this, $C_6$ contains only variables $h$ and $j$, both of which are also present in $C_7$. Since $C_6$ is a non-maximal clique, it is removed and its neighbour $C_4$ is connected to $C_7$. In $C_7$, $j$ is a non-interface variable, present in a single clique. We can follow a similar process of marginalization and removal of a non-maximal clique, leaving only $C_1, C_2, C_3, C_4$ and $C_5$ in $CTF_{1,a}$. We can further reduce the number of non-interface variables. Variable $i$ is present in cliques $C_4$ and $C_5$ which when collapsed give a clique of size 3 ($\leq mcs_{im}$) containing variables $h, i$ and $l$. Variable $i$ is removed and the beliefs are marginalized to give a new clique $C_4'$. Exact marginalization over all other NIVs will increase the clique size beyond $mcs_{im}$ and is therefore not attempted.

**Local marginalization:** In this step, we reduce clique sizes by removing variables from cliques with size greater than $mcs_{im}$ and locally marginalizing clique beliefs as follows. If a variable $v$ is locally marginalized from two adjacent cliques $C_i$ and $C_j$ with sepset $S_{i,j}$, the result is two cliques $C_i' = C_i \setminus v$ and $C_j' = C_j \setminus v$ with sepset $S_{i,j}' = S_{i,j} \setminus v$ and beliefs given by

$$\beta(C_i') = \sum_{D_v} \beta(C_i), \quad \beta(C_j') = \sum_{D_v} \beta(C_j), \quad \mu(S_{i,j}') = \sum_{D_v} \mu(S_{i,j}) \tag{5}$$

We need to ensure that local marginalization satisfies the following constraints:

(a) Since IVs are present in factors that have not yet been added to a CTF, they must be retained in at least one clique in $CTF_{k,a}$.

(b) A connected CT in $CTF_k$ should remain connected in $CTF_{k,a}$. The reason for this will become apparent in Section 5.

Figure 5 illustrates the methodology for local marginalization using the running example. $CTF_{1,a}$ obtained after exact marginalization contains a single clique $C_2$, with size greater than $mcs_{im}$ (set to 3). The variables present in this clique are $f, d, h$ and $m$. Since $f$ is an interface variable that is present in a single clique, it is not considered for marginalization. Variable $h$ is also not considered, because removal of $h$ from cliques $C_2$ and $C_4'$ will disconnect the clique tree, since the sepset between them contains only $h$. If we remove $d$ from $C_2$, it must also be removed from either $C_1$ or $C_3$ to satisfy RIP. We retain $d$ in $C_3$ and marginalize it from beliefs corresponding to $C_1$ and $C_2$. The resulting approximated CTF, $CTF_{1,a}$, contains cliques with sizes bounded by $mcs_{im}$.

### 4.3.1 Approximation Algorithm

*ApproximateCTF* (Algorithm 2) shows the formal steps in our algorithm used to approximate the CTF. The inputs are $CTF_k$, the set of factors $\Phi$ that have not been added to any CTF in the set $\{CTF_1, \ldots CTF_k\}$ and the clique size bound for the approximate CTF, $mcs_{im}$. It returns the approximate CTF, $CTF_{k,a}$. We first identify the interface variables ($IV$) and initialize $CTF_{k,a}$ as the minimal subgraph of $CTF_k$ that is needed to compute the joint beliefs of $IV$ ($MSG[IV]$, Definition 11) (lines 1-3). This is followed by exact marginalization of NIVs which are either present in a single clique or wherever the size of the collapsed clique is less than $mcs_{im}$ (lines 4-10). Next, we perform local marginalization to reduce clique sizes to $mcs_{im}$, if possible. We first choose a variable ($v$) that is present in large sized cliques and retain it in a connected subtree ($ST_r$) that has clique sizes less than or equal to $mcs_{im}$ (lines 12-16). $v$ is locally marginalized from all other cliques while satisfying the constraints specified for local marginalization (lines 18-24). Any non-maximal clique obtained after exact or local marginalization is removed and its neighbors are reconnected to the containing clique (lines 7,19).

### 4.3.2 Properties of the approximated CTF

If the input CTF, $CTF_k$ to the approximation algorithm is valid and calibrated, then resulting approximate CTF, $CTF_{k,a}$, satisfies the following properties. The proofs for these properties are included in Appendix A.

**Proposition 5.** All CTs in the approximate CTF, $CTF_{k,a}$, are valid CTs.

**Proposition 6.** All CTs in the approximate CTF, $CTF_{k,a}$, are calibrated.

**Proposition 7.** The normalization constant of all CTs in the approximate CTF $CTF_{k,a}$ is the same as in the input CTF, $CTF_k$.

**Proposition 8.** If the clique beliefs are uniform, then the beliefs obtained after local marginalization are exact.

### 4.3.3 Heuristics for choice of variables for local marginalization

Since our aim is to preserve the joint beliefs of the interface variables as much as possible, we would like to choose variables that have the least impact on this joint belief for local marginalization. We need a

---

**Algorithm 2** ApproximateCTF ($CTF_k$, $\Phi$, $mcs_{im}$)

---

**Input:** $CTF_k$: Input CTF

       $\Phi$: Set of remaining factors

       $mcs_{im}$: Maximum clique size limit for the approximated $CTF$

**Output:** $CTF_{k,a}$: Approximated $CTF$

 1: $IV \leftarrow \{Variables \in CTF_k\} \cap \{Variables \in \Phi\}$           ▷ Identify interface variables in $CTF_k$

 2: ▷ Initialize $CTF_{k,a}$

 3: $CTF_{k,a} \leftarrow$ Minimal subgraph of $CTF_k$ corresponding to variables in $IV$, $MSG[IV]$     ▷ See Definition 11

 4: $NIV \leftarrow$ Variables $\in CTF_{k,a} \setminus IV$         ▷ Identify non-interface variables in $CTF_{k,a}$

 5: ▷ Step1: Exact marginalization

 6: Sum out NIVs present in a single clique in $CTF_{k,a}$

 7: Remove resultant non-maximal cliques, re-connect neighbors

 8: **while** size of collapsed cliques $\leq mcs_{im}$ **do**     ▷ Exact marginalization over NIVs present in multiple cliques

 9:     Exact marginalization over NIVs; reconnect neighbors

10: **end while**

11: ▷ Step 2: Local marginalization

12: $L \leftarrow$ List of variables present in cliques with size $> mcs_{im}$     ▷ Identify variables present in large-sized cliques

13: ▷ Loop until max-clique size is $\leq mcs_{im}$ or further reduction is not possible

14: **while** ($CTF_{k,a}.max\text{-}clique\text{-}size > mcs_{im}$) && $L.isNotEmpty()$ **do**

15:     $v \leftarrow$ Choose a variable in $L$; prioritize NIVs

16:     $L.remove(v)$

17:     $ST_r \leftarrow$ Find a connected subtree containing $v$ s.t. max-clique-size $\leq mcs_{im}$  ▷ Subtree in which $v$ is retained

18:     $CTF'_{k,a} \leftarrow$ Locally marginalize $v$ from cliques and sepsets in $CTF_{k,a} \setminus ST_r$ ▷ Marginalize $v$ from other cliques

19:     Remove resultant non-maximal cliques, reconnect neighbors

20:     **if** (Any CT $\in CTF_{k,a}$ gets disconnected in $CTF'_{k,a}$) $\|$ (($v \in IV$)&&($v \notin CTF'_{k,a}$)) **then**

21:         continue     ▷ Ignore if a CT gets disconnected or if an IV is not retained after local marginalization

22:     **else**

23:         $CTF_{k,a} = CTF'_{k,a}$         ▷ Modify the approximate CTF

24:     **end if**

25: **end while**

26: **return** $CTF_{k,a}$

---

metric that measures this influence and is inexpensive to compute. Towards this end, we propose a heuristic technique based on pairwise mutual information (MI) between variables. The MI between two variables $x$ and $y$ is defined as

$$MI(x;y) = \sum_{s \in D_x, w \in D_y} p(s,w) \log \frac{p(s,w)}{p(s)p(w)}$$

Computation of MI for variables belonging to different cliques is expensive. Instead, we propose two metrics that are easy to compute, namely, *Maximum Local Mutual Information (MLMI)* and *Maximum Mutual Information (maxMI)* which are defined as follows. Let $IV_C$ denote the set of interface variables in a clique $C$. The $MLMI$ of a variable $v$ in clique $C$ is defined as

$$MLMI_{v,C} = \max_{\forall x \in IV_C \setminus v} MI(v;x) \tag{6}$$

The $maxMI$ for a variable $v$ is defined as the maximum $MLMI$ over all cliques.

$$maxMI_v = \max_{\forall C \in CTF \ s.t. \ v \in C} MLMI_{v,C} \tag{7}$$

As seen in Equation 6, if $v$ is an interface variable, $MLMI$ is the maximum MI between $v$ and the other interface variables in the clique. If $v$ is a non-interface variable, it is the maximum MI between $v$ and all the interface variables in the clique. Since $maxMI$ of $v$ is the maximum $MLMI$ over all cliques (Equation 7), it is a measure of the maximum influence that a variable $v$ has on interface variables that are present in cliques that contain $v$. A low $maxMI$ means that $v$ has a low $MI$ with interface variables in all the cliques in

which it is present and is therefore assumed to have a lower impact on the joint distribution of the interface variables.

We prioritize non-interface variables with the least $maxMI$ for local marginalization. If it is not possible to reduce clique sizes by removing non-interface variables, we locally marginalize over interface variables with least $maxMI$ (line 15, Algorithm 2). During local marginalization, if we find multiple connected subtrees ($ST_r$) with bounded clique sizes (line 17, Algorithm 2), we retain the variable in the subtree that contains the clique with the maximum $MLMI$.

### 4.3.4   Re-parameterization of approximate CTF

$CTF_{k+1}$ is constructed by adding new factors to the approximate CTF, $CTF_{k,a}$. Before adding new factors, we re-assign factors associated with cliques in $CTF_{k,a}$ such the product of these factors is a valid joint distribution. This reparameterization is needed to use the message-passing algorithm for calibration of $CTF_{k+1}$. Using Proposition 6, we know that clique and sepset beliefs in $CTF_{k,a}$ are calibrated. We re-assign clique factors as follows. For each CT in the $CTF_{k,a}$, a root node is chosen at random. The factor for the root node is the same as the clique belief. All other nodes are assigned factors by iterating through them in pre-order, i.e., from the root node to the leaf nodes. An un-visited neighbor $C_j{}'$ of a node $C_i{}'$ in $CTF_{k,a}$ is assigned the conditional belief $\beta(C_j{}'|C_i{}') = \frac{\beta(C_j{}')}{\mu(S_{i,j}')}$ as a factor. Using Equation 2, the product of the re-assigned factors is a valid joint distribution.

## 5   Approximate inference of the partition function

The partition function can be inferred using Proposition 9 and Theorem 2 stated below. The proofs for both are included in Appendix A.

**Proposition 9.** Let the undirected graph associated with the PGM be connected and let $\{CTF_1, CTF_{1,a}, CTF_2, \ldots, CTF_{n-1,a}, CTF_n\}$ with the corresponding sets of variables $\{X_1, X_{1,a}, X_2 \ldots X_{n-1,a}, X_n\}$ denote the sequence of CTFs generated by Algorithms 1 and 2. Then, the normalization constant of the distribution encoded by $CTF_k$ ($Z_k$) is

$$Z_k = \begin{cases} \sum\limits_{Domain(X_1)} \prod\limits_{\phi \in \Phi_1} \phi & \text{for } k = 1 \\ \sum\limits_{Domain(X_k)} \dfrac{\prod_{C' \in CTF_{k-1,a}} \beta(C')}{\prod_{SP' \in CTF_{k-1,a}} \mu(SP')} \prod\limits_{\phi \in \Phi_k} \phi & \text{for } k > 1 \end{cases} \tag{8}$$

where, $\Phi_1, \ldots, \Phi_k$ are the subsets of initial factors added to $CTF_1, \ldots CTF_k$ respectively and

$$\sum_{Domain(X_{k-1,a})} \frac{\prod_{C' \in CTF_{k-1,a}} \beta(C')}{\prod_{SP' \in CTF_{k-1,a}} \mu(SP')} = \sum_{Domain(X_{k-1})} \frac{\prod_{C \in CTF_{k-1}} \beta(C)}{\prod_{SP \in CTF_{k-1}} \mu(SP)} \tag{9}$$

**Theorem 2.** Let the undirected graph corresponding to the PGM be connected and let the sequence $\{CTF_1, \cdots, CTF_n\}$ be the SCTF generated by IBIA. Then, the last CTF, $CTF_n$ contains a single CT, denoted as $CT_n$. IBIA returns the normalization constant of $CT_n$ ($Z_n$) as the PR.

The PR returned by IBIA ($Z_n$) is an approximation to the exact value. This is because Algorithm 2 uses local marginalization, which means,

$$\frac{\prod_{C' \in CTF_{k-1,a}} \beta(C')}{\prod_{SP' \in CTF_{k-1,a}} \mu(SP')} \approx \sum_{Domain(X_{k-1} \setminus X_{k-1,a})} \frac{\prod_{C \in CTF_{k-1}} \beta(C)}{\prod_{SP \in CTF_{k-1}} \mu(SP)}$$

Note that, although the overall joint distribution in $CTF_{k-1,a}$ is approximate, the normalization constant is preserved as seen from Equation 9.

Evidence-based simplification of the PGM could give a set of disjoint graphs. We construct an SCTF corresponding to each connected graph. The PR is then estimated as the product of the normalization constants of the CT in the last CTF of each SCTF.

## 6 Complexity analysis and Conditions for exact inference

**Complexity**: Let $N_{CTF}$ be the number of CTFs in the SCTF and $N_s$ be the maximum number of incremental steps required to build any $CTF$. We now discuss the worst-case complexity of three steps used to construct the SCTF.

*Incremental Build*: In each step, we add a subset of factors that impact overlapping portions of the CTF. The overall complexity of modification depends on the number of steps and the cost of re-triangulation in each step. In the worst case, in each step we get a group of factors that impacts all the cliques in the CTF and there are no retained cliques. The cost of re-triangulation ($Cost_R$) using any of the greedy search methods is polynomial in the number of variables in CTF (Koller & Friedman, 2009, Chap. 9). Hence, the worst-case complexity is upper bounded by $O(N_{CTF} \cdot N_s \cdot Cost_R)$. Generally, the number of computations required is much lower since there are many retained cliques and different subsets of factors impact disjoint subgraphs of the existing CTs.

*Inference and Approximation:* Since we use exact inference to calibrate the clique-tree, the complexity of inference in each CTF is $O(2^{mcs_p})$. Approximation involves summing out variables from a belief table. Once again, this is $O(2^{mcs_p})$. The overall complexity is therefore $O(N_{CTF} \cdot 2^{mcs_p})$.

**Conditions under which IBIA gives exact solution:** When the SCTF has a single CTF, the PR obtained is exact. If the SCTF has multiple CTFs, it is still possible to get the exact PR if the approximate step uses only exact marginalization. But this is rare and in most cases, local marginalization is required, and the PR obtained is approximate.

## 7 Results

All experiments were carried out on a Intel i9-12900 Linux system. IBIA was run using Python v3.10 with Numpy, Scipy and NetworkX libraries. The memory limit was set to 8GB for all experiments, which is the same as that used in the UAI 2022 inference competition (UAI, 2022).

We address the following questions in our evaluation.

- How many instances can IBIA solve within different runtime limits?

- Are clique sizes generated by the proposed incremental method comparable to those obtained with a non-incremental method?

- Is the heuristic used for approximation useful?

- What is the impact of clique size constraints on the performance of IBIA?

- How does the performance of IBIA compare with the state of art techniques?

**Performance measure:** The error metric used is the absolute error in partition function ($PR$) measured as $|\log_{10} PR_{IBIA} - \log_{10} PR_{ref}|$. $PR_{ref}$ is either the exact value or available reference values of PR, discussed in more detail later in the section. Since each tool reports PR using a different number of precision digits, we round off errors to three decimal places and report an error of zero when it is less than 0.001.

**Benchmarks:** Table 1 lists the benchmark sets used in this work. These benchmarks have been included in several UAI approximate inference challenges (UAI, 2010; 2014; 2022) and the Probabilistic Inference Challenge (PIC, 2011). We have categorized an instance as *'small'* if the exact solution was either available in the repository (Ihler, 2006) or could be computed using Ace (Chavira & Darwiche, 2015; 2008), a tool based on weighted model counting. All other instances are categorized as *'large'*.

**Notation:** In all tables in this section, we denote the induced width of a specific benchmark as $w$ and the maximum domain-size as $dm$. We use the following to denote the average statistics over all instances in each benchmark set (a) $v_a$: average number of variables (b) $f_a$: average number of factors (c) $w_a$: average induced width and (d) $dm_a$: average of the maximum domain size.

**Choice of parameters:** Based on the memory limit of 8GB, we chose $mcs_p$ of 20 for all experiments unless stated otherwise. Since $mcs_{im}$ determines the extent of approximation, we would like it to be as high as possible for better accuracy. But, we also need a sufficient margin to add variables to the next CTF in the sequence. We have empirically chosen $mcs_{im}$ to be 5 less than $mcs_p$.

## 7.1 Number of instances solved by IBIA

Table 1: Statistics of benchmark sets used and percentage of total instances solved by IBIA with memory limit set to 8GB and runtime limit set to 20 seconds, 20 minutes, 60 minutes and 100 minutes.

| Size | Benchmarks | #Inst | Average stats [+] | Instances solved (%) | | | |
|---|---|---|---|---|---|---|---|
| | | | $(v_a, f_a, w_a, dm_a)$ | 20 s | 20 min | 60 min | 100 min |
| | Segmentation | 50 | (229,851,17,2) | 100% | 100% | 100% | 100% |
| | Promedas | 65 | (619,619,21,2) | 100% | 100% | 100% | 100% |
| | Protein | 77 | (60,180,6,76) | 100% | 100% | 100% | 100% |
| | BN | 97 | (637,637,28,10) | 100% | 100% | 100% | 100% |
| | Object Detection | 79 | (60,210,6,16) | 100% | 100% | 100% | 100% |
| *Small* | Grids | 8 | (250,728,22,2) | 100% | 100% | 100% | 100% |
| | CSP | 14 | (68,345,13,4) | 100% | 100% | 100% | 100% |
| | DBN | 66 | (780,15453,29,2) | 100% | 100% | 100% | 100% |
| | Pedigree | 24 | (853,853,24,5) | 100% | 100% | 100% | 100% |
| | mastermind | 128 | (2159,2159,26,2) | 98% | 100% | 100% | 100% |
| | blockmap | 240 | (24589,24589,5057,2) | 78% | 100% | 100% | 100% |
| | Segmentation | 50 | (229,851,19,21) | 100% | 100% | 100% | 100% |
| | Promedas | 173 | (1209,1209,72,2) | 80% | 100% | 100% | 100% |
| | Protein | 386 | (311,1215,21,81) | 75% | 100% | 100% | 100% |
| | BN | 22 | (1272,1272,51,17) | 73% | 100% | 100% | 100% |
| *Large* | Object Detection | 37 | (60,1830,59,17) | 0% | 100% | 100% | 100% |
| | Grids | 19 | (3432,10244,117,2) | 16% | 79% | 100% | 100% |
| | CSP | 52 | (304,12168,181,43) | 23% | 77% | 77% | 77% |
| | DBN | 48 | (1000,66116,78,2) | 0% | 63% | 63% | 100% |
| | Type4b | 82 | (10822,10822,24,5) | 0% | 99% | 100% | 100% |

[+] Average statistics for instances in each benchmark set, $v_a$: average number of variables, $f_a$: average number of factors, $w_a$: average induced width and $dm_a$: average of the maximum domain-size.

Table 1 shows the percentage of large and small instances in each set that are solved by IBIA within 20 seconds, 20 minutes, 60 minutes and 100 minutes, similar to limits used in the UAI 2022 competition.

Except for a few blockmap and some mastermind instances, IBIA was able to solve all the small benchmarks within 20 seconds. Solutions to the remaining instances were obtained within 20 minutes. For the large instances, we allow for an increase in $mcs_p$ if needed so that at least one new factor can be added while maintaining the overall memory limit. Except for Grids, CSP, DBN and Type4b, in which some instances take longer, all other large instances could be solved within 20 minutes. All Grids and Type4b instances can be solved within 60 minutes and DBN within 100 minutes. For a few DBN instances, the number of factors is very large (greater than 100,000) and the runtime is dominated by the incremental build step where repeated re-triangulations are performed to add factors. In large CSP benchmarks, inference using IBIA runs out of memory in 12 out of 52 instances. For these instances, the maximum domain-size is large (varies from 44 to 200). As a result, the number of variables contained in cliques and sepsets in the CTF is very small. Therefore, the approximation step has a limited choice of variables and it becomes infeasible in these cases. This in turn leads to large-sized cliques in the next CTF, thereby exceeding the set memory limit.

## 7.2 Evaluation of Algorithms in IBIA

In this section, we evaluate the performance of the proposed method for incremental CT construction and the performance of the metric used for guiding the approximate step in IBIA. We also study the trade-off between runtime and accuracy.

**Evaluation of Incremental CT construction:** We first evaluated our algorithm for incremental construction of the CT in terms of the maximum clique size. We used the following method for evaluation. For a given $mcs_p$, we used Algorithm 1 to incrementally construct the first CTF in the sequence ($CTF_1$). For comparison, we used a CTF obtained using full compilation of all the factors added to $CTF_1$. This is done as follows. We first find the undirected graph induced by the factors that were added to $CTF_1$. This

graph is then compiled using variable elimination (Zhang & Poole, 1996; Koller & Friedman, 2009). The elimination order is found using the 'min-fill' metric, and the metric 'min-neighbors' is used in the case of a tie (Koller & Friedman, 2009). We choose the min-fill metric since in most cases it has found to give lower clique sizes than other heuristics (Gogate & Dechter, 2004; Li & Ueno, 2017). Re-computing the number of fill-in edges each time a variable is eliminated increases the execution time. Therefore, we adopted the methodology suggested in Kask et al. (2011) to compute only the change in the number of fill-in edges.

Table 2: The difference in maximum clique sizes obtained after incremental construction ($mcs_{ibia}$) of the first CTF in the sequence, $CTF_1$, and that obtained after full compilation of undirected graph induced by the factors added to $CTF_1$ ($mcs_f$) for $mcs_p = 20, 25$. $\Delta = mcs_{ibia} - mcs_f$.

| | #Inst | $dm_a^+$ | $mcs_p = 20$ | | | $mcs_p = 25$ | | |
|---|---|---|---|---|---|---|---|---|
| | | | Avg $\Delta$ | Min $\Delta$ | Max $\Delta$ | Avg $\Delta$ | Min $\Delta$ | Max $\Delta$ |
| BN | 119 | 12 | -0.03 | -8.1 | 3 | 0.5 | -9.2 | 5 |
| Promedas | 238 | 2 | -0.6 | -12 | 4 | -0.1 | -14 | 6 |
| Pedigree | 24 | 5 | -2.1 | -11.8 | 3 | -1.5 | -10.3 | 3 |
| Grids | 27 | 2 | -0.6 | -3 | 2 | -0.2 | -6 | 6 |
| CSP | 66 | 35 | -1.5 | -13.3 | 4 | -2 | -14.3 | 3.6 |

$^+$ $dm_a$: average of the maximum domain-size.

Table 2 compares the maximum clique size obtained using the incremental ($mcs_{ibia}$) and full compilation ($mcs_f$) approaches for $mcs_p$ of 20 and 25. It shows the average, maximum and minimum difference ($\Delta = mcs_{ibia} - mcs_f$) in clique sizes[1] for a few benchmark sets. The results for other benchmarks are similar. The difference, $\Delta = mcs_{ibia} - mcs_f$, is negative when the incremental approach yields a smaller clique size and positive otherwise. On an average, our incremental approach gives similar results as full compilation of the corresponding undirected graph. The average is negative, indicating that in many benchmarks, the incremental approach actually resulted in lower clique sizes than full compilation. Since the maximum value of $\Delta$ is positive, it indicates that there are instances for which full compilation is better, which is expected.

**Evaluation of heuristic used in the approximate step:** To get an approximate CTF with lower clique sizes, we choose variables for local marginalization based on the $maxMI$ metric (refer Equation 7). Table 3 compares the errors obtained using the $maxMI$ metric and errors obtained using a random selection of variables. The minimum error obtained is marked in bold. We show results for a subset of hard instances (large width and domain-sizes) in BN, Pedigree, Promedas and DBN benchmarks. In most of the testcases, we observe that the errors obtained with the $maxMI$ metric are either lower or comparable to that obtained using a random selection. This shows that the metric performs well.

Table 3: Comparison of error obtained using IBIA when the choice of variables for local marginalization is made based on the $maxMI$ metric versus a random selection of variables. The minimum error obtained is marked in bold.

| Benchmark | $(w, dm)^+$ | Error | | Benchmark | $(w, dm)^+$ | Error | |
|---|---|---|---|---|---|---|---|
| | | $maxMI$ | Random | | | $maxMI$ | Random |
| BN_69 | (48,36) | **1.2** | 1.3 | or_chain_155 | (31,2) | **0.01** | 0.02 |
| BN_70 | (81,36) | **2.2** | 5.1 | or_chain_107 | (33,2) | **0.3** | **0.3** |
| BN_71 | (45,36) | **0.8** | 2.2 | or_chain_128 | (30,2) | **0.2** | 0.6 |
| BN_72 | (58,36) | **1.3** | 2.4 | or_chain_102 | (31,2) | **0.4** | 0.8 |
| BN_73 | (75,36) | **1.9** | 2.3 | or_chain_106 | (31,2) | **0.3** | 0.7 |
| BN_74 | (37,36) | **1.7** | 2.9 | or_chain_140 | (33,2) | **0.1** | 0.9 |
| BN_75 | (59,36) | **2.4** | 2.5 | or_chain_242 | (31,2) | 0.5 | **0.1** |
| BN_76 | (53,36) | **1.7** | **1.7** | or_chain_198 | (32,2) | 1.0 | **0.01** |
| pedigree13 | (32,3) | **0.01** | 0.02 | or_chain_61 | (34,2) | 0.6 | **0.1** |
| pedigree42 | (24,5) | 0.05 | **0.04** | rus_20_40_0_3 | (30,2) | **0.9** | 2.6 |
| pedigree19 | (27,5) | **0.04** | 0.3 | rus2_20_40_2_2 | (30,2) | **0.4** | 1.1 |
| pedigree34 | (32,5) | **0.2** | 0.3 | rus2_20_40_8_2 | (30,2) | **0.7** | 1.3 |
| pedigree40 | (29,7) | **0.1** | 0.3 | rus_20_40_4_2 | (30,2) | 0.4 | **0.02** |
| pedigree41 | (31,5) | **0.04** | 0.5 | rus_20_40_8_1 | (30,2) | 0.8 | **0.2** |
| pedigree7 | (33,4) | **0.01** | 0.2 | rus2_20_40_5_3 | (30,2) | 0.7 | **0.3** |

$^+$ $w$ : induced width, $dm$ : maximum domain-size

---

[1]As shown in Equation (3), our definition for clique size is the logarithm (base 2) of the product of the domain sizes. Therefore, it is possible to get decimal values for sizes when cliques contain variables with domain size greater than 2.

**Impact of $mcs_p$ on accuracy and runtime:** Table 4 shows the error in the estimated PR values for various values of $mcs_p$. As mentioned earlier, we have empirically chosen $mcs_{im}$ to be 5 less than $mcs_p$. We observe that in most cases the accuracy improves as the clique size bounds are increased. This is expected because increasing the bounds potentially increases the number of factors added in each step, which in turn could reduce the number of CTFs and the number of approximate steps. Also, the metrics used for guiding the approximate step are computed using beliefs corresponding to the partial set of factors added up to the current CTF. Therefore, the accuracy of the metrics could improve when a larger set of factors is added, resulting in better estimates.

Table 4: Comparison of error in partition function estimated with IBIA and required runtime (in seconds) for various clique size constraints $(mcs_p, mcs_{im})$.

| Benchmark | $(w, dm)^+$ | Error | | | | Runtime (s) | | | |
|---|---|---|---|---|---|---|---|---|---|
| | | (10,5) | (15,10) | (20,15) | (25,20) | (10,5) | (15,10) | (20,15) | (25,20) |
| grid2020f15 | (26,2) | 9.7 | 2.9 | 0.5 | $4\times10^{-6}$ | 3 | 4 | 3 | 28 |
| grid2020f2 | (26,2) | 0.1 | 0.1 | $3\times10^{-5}$ | $3\times10^{-6}$ | 3 | 4 | 3 | 29 |
| grid2020f5 | (26,2) | 0.2 | 0.2 | 0.003 | $3\times10^{-6}$ | 3 | 4 | 3 | 34 |
| rus2_20_40_3_2 | (30,2) | 3.4 | 2.1 | 0.9 | 0.1 | 17 | 11 | 13 | 261 |
| rus2_20_40_2_2 | (30,2) | 1.8 | 1.7 | 0.4 | 0.9 | 16 | 12 | 13 | 261 |
| rus2_20_40_2_3 | (30,2) | 5.6 | 0.7 | 1.4 | 0.01 | 18 | 14 | 14 | 165 |
| rus2_20_40_6_2 | (30,2) | 6.1 | 1.3 | 1.0 | 0.2 | 17 | 12 | 13 | 185 |
| pedigree19 | (27,5) | 0.6 | 0.4 | 0.04 | 0.01 | 3 | 3 | 4 | 36 |
| pedigree31 | (29,5) | 0.2 | 0.1 | 0.1 | 0.01 | 5 | 4 | 5 | 45 |
| pedigree34 | (32,5) | 0.4 | 0.2 | 0.2 | 0.08 | 3 | 3 | 4 | 35 |
| pedigree40 | (29,7) | 0.8 | 0.4 | 0.1 | 0.05 | 5 | 5 | 5 | 44 |
| pedigree42 | (24,5) | 0.1 | 0.1 | 0.04 | 0.004 | 1 | 1 | 1 | 19 |
| pedigree44 | (27,4) | 0.3 | 0.1 | 0.03 | 0.001 | 3 | 2 | 3 | 19 |

$^+$ $w$: induced width, $dm$: maximum domain-size

The runtime of IBIA includes the time required for the construction of the SCTF and inference of the partition function. We observe that while the required runtime is similar when $mcs_p$ is set to $10, 15$ and $20$, it increases sharply when $mcs_p$ is set to 25. This is because the build step dominates the runtime for smaller values of $mcs_p$ and the infer step dominates for larger values. As discussed in Section 6, the time complexity of the build step is $O(N_{CTF} \cdot N_s \cdot Cost_R)$. As $mcs_p$ increases, while the number of CTFs in the sequence ($N_{CTF}$) is expected to reduce, the cost of re-triangulation ($Cost_R$) could be potentially larger as the number of variables in the CTF is larger. Therefore, we observe that the runtime is similar for $mcs_p$ of $10, 15$ and $20$. The exponential complexity of inference begins to dominate at $mcs_p = 25$.

## 7.3 Accuracy and runtime comparison with existing inference techniques

### 7.3.1 Methods used for comparison

As mentioned, we classified the benchmarks as small or large depending on whether exact PR values can be computed or not. To evaluate the performance of IBIA for the small benchmarks, we used the results of a recent evaluation of various exact and approximate inference solvers by Agrawal et al. (2021). Based on these results, we chose the following methods for comparison. For exact inference, we used Ace (Chavira & Darwiche, 2015), which is based on weighted model counting. To compare with variational methods, we used LBP (Murphy et al., 1999) and double-loop GBP (HAK) (Heskes et al., 2003). Amongst the sampling techniques with a variational proposal, we chose Sample search (Gogate & Dechter, 2011). We used the publicly available codes used in Agrawal et al. (2021) or original implementations by the authors of the method for the comparison. Accordingly, for LBP and HAK, we used the implementations in libDAI (Mooij, 2010). For SampleSearch, we used a recent implementation (Gogate, 2020) by the authors of the method, which performs sample search using an IJGP-based proposal and cutset sampling (ISSwc). The runtime switches used are included in Table 5. For IBIA, we have used two sets of clique size bounds. We refer to IBIA with $mcs_p$ set to 20 as *'IBIA20'* and IBIA with $mcs_p$ of 25 as *'IBIA25'*. We report results for ISSwc with two parameter settings. The first variant called as *'ISSwcd'* uses default values of *ibound* (effective number of binary variables in a cluster) and w-cutset bound determined by the solver depending on the benchmark and given runtime constraints. For a fair comparison with IBIA, we set both bounds to 20 in the second variant (referred to as *'ISSwc20'*). While IBIA is implemented in Python, other tools use C++.

Table 5: Methods used for comparison. For each method, we indicate the class of techniques it falls under. The column marked Publication has the citation to the paper containing the estimates of the PR and the first column has the citation to the original paper of the method. Methods for which we obtained data by running various tools are shown with the corresponding parameter settings in the last two columns.

| Method | Type | Publication | Tool | Parameters |
|---|---|---|---|---|
| IBIA | | | IBIA | $mcs_p = 20, mcs_{im} = 15$ (IBIA20) |
| | | | | $mcs_p = 25, mcs_{im} = 20$ (IBIA25) |
| LBP (Murphy et al., 1999) | Variational | | LibDAI | $tol = 10^{-3}$,#Iter= $10^4$ |
| HAK (Heskes et al., 2003) | Variational | | LibDAI | $tol = 10^{-3}$,#Iter= $10^4$, clusters=LOOP3 |
| ISSwc (Gogate & Dechter, 2011) | Variational (MB) +Sampling | ✓(Gogate & Dechter, 2011) | ISSwc | Default (ISSwcd) ibound=20,w-cutset bound=20 (ISSwc20) |
| EDBP (Choi & Darwiche, 2006) | Variational | ✓(Gogate & Dechter, 2011) | | |
| WMB (Liu & Ihler, 2011) | Variational (MB) | ✓(Agarwal et al., 2022) | | |
| NeuroBE (Agarwal et al., 2022) | Variational (MB) +Neural Networks | ✓(Agarwal et al., 2022) | | |
| DBE (Razeghi et al., 2021) | Variational (MB) + Neural Networks | ✓(Razeghi et al., 2021) | | |
| DIS (Lou et al., 2017; 2019) | Variational (MB) +Search +Sampling | ✓(Kask et al., 2020) | | |
| AS (Broka, 2018) | Variational (MB) +Search+Sampling | ✓(Kask et al., 2020) | | |

Amongst the small benchmarks, some of the benchmarks are in general considered "hard" in the literature. These benchmarks have been used extensively for comparison and results for many approximate inference methods are available in the literature. For these benchmarks, we compared our method with published results. Table 5 has the methods used for comparison and the reference to the publication from which the PR estimates were obtained.

For large networks for which the exact PR is not available, we compare our results with published results in Kask et al. (2020), which uses reference values of PR generated using 100 1-hr runs of abstraction sampling.

### 7.3.2 Performance of IBIA for the small benchmarks

Table 6 compares the average error obtained using IBIA20 and IBIA25 with LBP, HAK, ISSwcd and ISSwc20 for all small benchmarks. We use two runtime constraints, 20 seconds and 20 minutes. If all instances in a set could not be solved within the given time and memory limits, we mark the corresponding entry as '-' and show the number of instances solved in brackets. An entry is marked in bold if it gives the lowest error amongst the methods used for comparison.

Out of 848 instances, IBIA20 solves 792 instances in 20 seconds. In contrast, ISSwc20 that uses the same clique size bounds solves only 659 instances. ISSwcd uses smaller clique size constraints and is able to solve 838 instances. LBP and HAK solve lesser instances than IBIA20. Note that while other solvers are written in C++, IBIA is implemented using Python3 and is therefore at a disadvantage in terms of runtime. That said, the only benchmarks that do not run within 20 seconds with IBIA20 are blockmap and mastermind, for which the maximum runtime is 408 and 35 seconds respectively. In 20 minutes, IBIA20 is able to solve all instances. On the other hand, LBP is unable to solve a few DBN instances, ISSwcd is unable to solve a few BN instances and ISSwc20 is unable to solve some Grid, BN and mastermind testcases. IBIA25 also fails to give a solution for some relational (blockmap and mastermind) and DBN benchmarks in 20 minutes with $8GB$ memory limit.

For both time constraints, IBIA20 is definitely better than the two variational methods LBP and HAK for all benchmark sets. In 20 seconds, the errors obtained using IBIA20 are comparable to or better than ISSwcd and ISSwc20 for all benchmarks except CSP. IBIA20 has a significantly lower error for Pedigree and Grids, but higher error than ISSwcd for CSP. It is the only solver that solves all BN benchmarks in 20 seconds, with a low error. In 20 minutes, the lowest errors are obtained by either ISSwcd or IBIA25 or both. **In fact, the accuracy of the PR estimates obtained with IBIA20 in 20 seconds is either comparable to or better than that obtained by ISSwc20 and ISSwcd in 20 minutes for many of the benchmark**

Table 6: Average error in partition function estimated using IBIA20, IBIA25, LBP, HAK, ISSwcd and ISSwc20 with runtime limit set to 20 seconds and 20 minutes. Entries are marked as '-' where at least one instance could not be solved within the set time limit and the number of instances solved is shown in brackets below. The minimum average error obtained for each set is marked in bold.

| | Average stats $(f_a, w_a, dm_a)^+$ | Total #Inst. | Average Error (20 seconds) (#Inst.) | | | | | | Average Error (20 minutes) (#Inst.) | | | | | |
|---|---|---|---|---|---|---|---|---|---|---|---|---|---|---|
| | | | LBP | HAK | ISSwcd | ISSwc20 | IBIA20 | IBIA25 | LBP | HAK | ISSwcd | ISSwc20 | IBIA20 | IBIA25 |
| Pedigree | (853,24,4) | 24 | - (22) | 1.03 | 2.48 | 0.41 | **0.07** | - (12) | 2.60 | 1.03 | 0.17 | 0.20 | 0.07 | **0.05** |
| Grids | (728,22,2) | 8 | 45.5 | 113.3 | 6.1 | - (4) | **0.2** | - (4) | 45.5 | 113.3 | 0.4 | - (4) | 0.2 | **0** |
| Promedas | (619,21,2) | 65 | **0.2** | **0.2** | 0.7 | - (62) | **0.2** | - (30) | 0.2 | 0.2 | **0.1** | **0.1** | 0.2 | **0.1** |
| DBN | (15453,29,2) | 66 | - (57) | - (63) | 0.82 | - (6) | **0.57** | - (6) | - (58) | 30.2 | **0.001** | 0.02 | 0.57 | - (36) |
| CSP | (345,13,4) | 14 | 18.2 | 12.5 | **0.68** | - (11) | 2.87 | - (11) | 18.2 | 12.5 | **0.28** | 0.43 | 2.87 | 1.06 |
| BN | (637,28,10) | 97 | - (84) | - (54) | - (91) | - (77) | **0.004** | - (84) | 0.27 | - (78) | - (93) | - (91) | 0.004 | **0.002** |
| ObjDetect | (210,6,16) | 79 | 0.38 | - (35) | 0.05 | - (22) | 0.01 | **0** | 0.38 | 6.2 | 0.01 | 0.001 | 0.01 | **0** |
| Protein | (180,6,76) | 77 | 0.004 | - (34) | 0.001 | 0.003 | **0** | **0** | 0.004 | 0.006 | **0** | **0** | **0** | **0** |
| Segment | (851,17,2) | 50 | 0.62 | 0.05 | 0.07 | - (46) | 0.001 | **0** | 0.62 | 0.05 | **0** | 0.005 | 0.001 | **0** |
| Blockmap | (24589,5057,2) | 240 | - (186) | - (75) | - (236) | - (236) | - (187) | - (177) | - (238) | - (152) | **0** | **0** | 0.009 | - (198) |
| Mastermind | (2159,26,2) | 128 | - (112) | - (103) | **0.13** | - (94) | - (125) | - (96) | 2.33 | 2.37 | **0.04** | - (113) | 0.12 | - (119) |
| Total #Instances solved | | 848 | 754 | 525 | 838 | 659 | 792 | 626 | 838 | 741 | 844 | 823 | 848 | 767 |

$^+ f_a$: average number of factors, $w_a$: average induced width and $dm_a$: average of the maximum domain-size.

**sets.** The hardest benchmarks for IBIA are CSP and DBN. In ISSwc, cutset sampling plays a crucial role in reduction of errors. Without cutset sampling, we found that errors are significantly larger. This is also seen from the results in Broka (2018).

### 7.3.3 Comparison with published results

Table 7 compares the error obtained using IBIA ($mcs_p = 10, 20, 25$) with WMB, DBE, NeuroBE, EDBP and ISSwc for five subsets of benchmarks. The memory limit for IBIA was set to 8GB and time limit to 20 minutes. In the table, we use ISSwc(P) to indicate that the reported results are published results. For fair comparison, we set $mcs_p$ to 10 in IBIA for benchmarks where *ibound* of 10 was used in published results. Entries are marked with '-' for instances where published results are not available for a particular benchmark. The minimum error obtained for each testcase is marked in magenta color in the table.

For small grid instances, the error obtained using IBIA10 is lower than all other methods. For small DBN instances, the error obtained with IBIA10 is smaller than WMB10, but worse than DBE10. For these instances, IBIA20 has the best accuracy in all testcases, except rbm20 for which WMB20 is better. In the Pedigree and BN instances, IBIA20 gives an error comparable to ISSwc(P). The two exceptions are BN_72 and BN_75 where IBIA20 gives a significantly larger error. For these instances, IBIA25 gives error comparable to ISSwc(P). Exact solutions are not known for large Grid instances. Therefore, we measure the absolute difference from the reference values published in Agarwal et al. (2022); Razeghi et al. (2021). The difference obtained with IBIA20 is much smaller than WMB20 and DBE20, and higher than NeuroBE for some instances. That said, the reference values are estimates and not the exact solution, thereby making it difficult to draw any conclusions.

Runtimes for published data cannot be compared due to differences in programming languages and systems used for evaluation. Therefore, we have only reported runtimes for IBIA. The small instances can be solved in less than 10 seconds by IBIA20. For the larger BNs and Grids, IBIA requires a couple of 100s to get an error comparable to ISSwc(P) and NeuroBE20 respectively.

Table 7: Comparison of error in PR obtained with IBIA with published results for a subset of benchmarks. The minimum error obtained for each testcase is shown in magenta. Entries are marked with '-' where published results are not available. $w$: induced width, $dm$: maximum domain-size

(a) Grid-small ($mcs_p = 10, ibound = 10$)

| | $(w, dm)$ | $\log_{10} PR$ | Error | | | | Runtime (s) |
|---|---|---|---|---|---|---|---|
| | | | WMB10 | DBE10 | NeuroBE10 | IBIA10 | IBIA10 |
| grid1010f10w | (21,2) | 333.3 | 32 | 4 | 1.8 | **0.05** | 0.5 |
| grid1010f10 | (12,2) | 303.1 | 1.6 | 0.7 | 1.2 | **0** | 0.2 |
| grid2020f2 | (26,2) | 291.7 | 11 | 2 | 0.1 | **0** | 3 |
| grid2020f10 | (26,2) | 1312.0 | 81 | 10 | 2.4 | **0.002** | 3 |
| grid2020f5 | (26,2) | 665.1 | 39 | 6 | 0.8 | **0.003** | 3 |
| grid2020f15 | (26,2) | 1963.0 | 123 | 18 | 2.7 | **0.5** | 3 |

(b) DBN-small ($mcs_p = 10, ibound = 10$ and $mcs_p = 20, ibound = 20$)

| | $(w, dm)$ | $\log_{10} PR$ | Error | | | | Runtime (s) |
|---|---|---|---|---|---|---|---|
| | | | WMB10 | DBE10 | NeuroBE10 | IBIA10 | IBIA10 |
| rbm20 | (20,2) | 58.5 | 7.8 | **0.5** | 1.4 | 2.1 | 1 |
| rbm21 | (21,2) | 63.1 | 15.7 | **0.7** | 0.9 | 1.0 | 3 |
| rbm22 | (22,2) | 66.6 | 27.5 | **0.7** | 0.9 | 6.0 | 3 |
| | | | WMB20 | DBE20 | NeuroBE20 | IBIA20 | IBIA20 |
| rbm20 | (20,2) | 58.5 | **0.0007** | 0.2 | 0.4 | 0.1 | 2 |
| rbm21 | (21,2) | 63.1 | 6.4 | 0.4 | 0.6 | **0.2** | 3 |
| rbm22 | (22,2) | 66.6 | 8.7 | 0.5 | 0.8 | **0.1** | 5 |
| rbm-ferro20 | (20,2) | 151.2 | 0.005 | 0.4 | 0.8 | **0** | 2 |
| rbm-ferro21 | (21,2) | 152.6 | 2.0 | 1.1 | 1.8 | **0** | 3 |
| rbm-ferro22 | (22,2) | 166.1 | 0.5 | 2.1 | 4.2 | **0** | 5 |

(c) Pedigree-small ($mcs_p = 20, ibound = 20$)

| | $(w, dm)$ | $\log_{10} PR$ | Error | | | | | | Runtime (s) |
|---|---|---|---|---|---|---|---|---|---|
| | | | EDBP | ISSwc(P)[1] | WMB20 | DBE20 | NeuroBE20 | IBIA20 | IBIA20 |
| pedigree19 | (27,5) | -59.0 | 0.5 | 0.14 | 2.6 | 3.7 | 2.6 | **0.04** | 4 |
| pedigree42 | (24,5) | -30.8 | 0.3 | **0** | - | - | - | 0.05 | 1 |
| pedigree44 | (27,4) | -63.5 | 1.6 | **0** | - | - | - | 0.04 | 3 |
| pedigree41 | (31,5) | -76.0 | - | - | 4.1 | 2.9 | 0.5 | **0.04** | 4 |
| pedigree31 | (29,5) | -69.7 | 0.2 | **0.02** | 12.4 | 5.9 | - | 0.1 | 5 |
| pedigree13 | (32,3) | -31.2 | 1.9 | 0.11 | 6.5 | 3.9 | 1.1 | **0.01** | 6 |
| pedigree34 | (32,5) | -64.2 | **0.1** | 0.2 | 7.1 | 5.9 | 0.7 | 0.2 | 4 |
| pedigree7 | (33,4) | -64.8 | 0.7 | 0.05 | 6.0 | 6.0 | 1.8 | **0.01** | 4 |

[1] Results for sample search with IJGP-based proposal and cutset sampling as published in Gogate & Dechter (2011)

(d) BN-large ($mcs_p = 20, 25$)

| | $(w, dm)$ | $\log_{10} PR^0$ | Error | | | | Runtime (s) | |
|---|---|---|---|---|---|---|---|---|
| | | | EDBP | ISSwc(P)[1] | IBIA20 | IBIA25 | IBIA20 | IBIA25 |
| BN_69 | (48,36) | -53.3 | 3.3 | 1.3 | **1.2** | **1.2** | 6 | 140 |
| BN_70 | (81,36) | -70.7 | 7.5 | 2.2 | 2.2 | **0.1** | 25 | 404 |
| BN_71 | (45,36) | -110.3 | 3.7 | **0.6** | 0.8 | 0.8 | 14 | 197 |
| BN_72 | (58,36) | -149.4 | 4.7 | **0.1** | 1.3 | 0.6 | 28 | 261 |
| BN_73 | (75,36) | -112.6 | 5.0 | 2.0 | 1.9 | **1.3** | 16 | 233 |
| BN_74 | (37,36) | -44.4 | 2.8 | 1.3 | 1.7 | **1.1** | 3 | 68 |
| BN_75 | (59,36) | -90.2 | 5.3 | **0.4** | 2.4 | 0.8 | 23 | 360 |
| BN_76 | (53,36) | -109.3 | 4.1 | 1.4 | 1.7 | **1.0** | 21 | 268 |

[0] Exact values computed using bucket elimination with external memory as published in Gogate & Dechter (2011)
[1] Results for sample search with IJGP-based proposal and cutset sampling as published in Gogate & Dechter (2011)

(e) Grid-large ($mcs_p = 20$)

| | $(w, dm)$ | $\log_{10} PR^0_{Ref}$ | Error | | | | Runtime (s) |
|---|---|---|---|---|---|---|---|
| | | | WMB20 | DBE20 | NeuroBE20 | IBIA20 | IBIA20 |
| grid4040f2 | (54,2) | 1220 | 25 | 7 | 2 | **0.2** | 30 |
| grid4040f5 | (54,2) | 2800 | 85 | 40 | 4 | **2** | 28 |
| grid4040f10 | (54,2) | 5490 | 215 | 97 | **10** | 21 | 30 |
| grid4040f15 | (54,2) | 8200 | 338 | 83 | **18** | 36 | 29 |
| grid4040f2w | (113,2) | 1231 | 32 | 15 | 6 | **0.6** | 144 |
| grid4040f5w | (113,2) | 2819 | 137 | - | **10** | 24 | 159 |
| grid4040f10w | (113,2) | 5637 | 298 | - | **54** | 75 | 140 |

[0] Reference values computed using $100 \times 1hr$ runs of abstraction sampling as published in Agarwal et al. (2022); Razeghi et al. (2021)

Table 8: Comparison of the average difference in PR from the reference values ($\log_{10} PR - \log_{10} PR_{ref}$) for large instances in four benchmark sets. $PR_{ref}$ are estimates averaged over $100 \times 1hr$ simulations of abstraction sampling (AS). Table reports results obtained using IBIA ($mcs_p = 20$) and published results for DIS and a single 1hr run of AS. Entries are marked as '-' where at least one instance could not be solved.
AS(R): AOAS with randomized context-based abstraction function with 256 levels
AS(B): Best configuration

| | #Inst. | Avg. Difference | | | | Avg. (Max.) Runtime (s) |
|---|---|---|---|---|---|---|
| | | IBIA20 | AS(R) | AS(B) | DIS | IBIA20 |
| Promedas | 173 | 1.0 | -3.5 | -2.8 | -66.6 | 13 (60) |
| Grids | 19 | 73.2 | -77.5 | -49.5 | -113.73 | 16 (26) |
| Type4b | 67[†] | 8.6 | - | - | - | 336 (1226) |
| DBN | 48 | -16.2 | -2.6 | -2.3 | -39.5 | 2000 (5810) |

[†] Reference values available only for 67 large instances out of 82.

Table 8 has the results for large benchmarks for which the exact PR is not known. Here, the comparison was done with reference values of PR.[2] The table reports the average difference over all instances in 4 benchmark sets. It has the results obtained using IBIA20 as well as published results for dynamic importance sampling (DIS) and abstraction sampling (AS) (Kask et al., 2020). In the table, the column marked AS(R) shows results obtained using the AOAS algorithm with randomized context-based abstraction function with 256 levels, and the column marked AS(B) shows best-case results. The table also shows the average and maximum runtime required for IBIA20.

IBIA20 could solve all instances within 8GB memory. In contrast, both AS and DIS are unable to solve all Type4b benchmarks within 1hr and 24 GB memory (Kask et al., 2020). On an average, estimates obtained using IBIA20 are higher than the reference value for all benchmarks except DBN, while those obtained by AS and DIS are lower. While Promedas, Grids and Type4b benchmarks are easy for IBIA, the DBN benchmarks are difficult. Both the difference from the reference and required runtimes are larger for these testcases.

## 8 Related work

**Inference methods that use multiple CTFs**: As discussed in Section 3.2, two issues need to be addressed in inference techniques that divide the PGM into multiple sections. The first issue is how to divide the PGM such that the maximum clique size of the CTF corresponding to each section is bounded. Previous attempts at dividing the PGM include the exact inference method, multiply sectioned BN (MSBN) (Xiang et al., 1993; Xiang & Lesser, 2003), and the approximate inference method in Bhanja & Ranganathan (2004). In MSBNs, the network structure is divided into sections. The CTF for each section is built using co-operative triangulation and there is no guarantee that the maximum clique size will be within a specified bound. Bhanja & Ranganathan (2004) use an iterative method for partitioning that requires repeated conversion of different candidate sections to corresponding CTFs, which is compute-intensive. In contrast, IBIA uses an incremental strategy for sectioning that requires re-triangulation of only a portion of the CTF in each step, which reduces the time complexity.

The second issue is how do we exchange beliefs between the CTFs so that the overall partition function can be inferred. In Bhanja & Ranganathan (2004), the information is transmitted from one CTF to the next using approximate Chow-Liu trees, containing variables present in both CTFs. However, this method cannot be used for inference of PR since finding a connected Chow-Liu tree consisting of all interface variables requires exact marginalization which is computationally infeasible. In contrast, information is passed via an approximate CTF in IBIA, which is constructed using a combination of exact and local marginalization. A sequence of CTs is also obtained for the 2T-BN model of dynamic BNs in the method proposed in Murphy (2002). Beliefs are transferred from one CT to the next using joint beliefs over subsets of variables present in both CTs (Boyen & Koller, 1998). The approximation method used in this technique could disconnect CTs and hence, cannot be used for inference of PR. In contrast, the approximation strategy used in IBIA

---

[2]Reference values of PR were obtained by Prof. Rina Dechter's group by averaging estimates obtained from 100 one hour runs of abstraction sampling.

preserves the PR at each step (Proposition 7). This allows for inference of the overall PR from the last CTF in the sequence.

**Incremental construction of CTs**: Incremental methods for CT modification have been explored in some previous works (Draper, 1995; Darwiche, 1998; Flores et al., 2002). In Draper (1995), incremental addition of links is performed by first forming a cluster graph using a set of rules and then converting the cluster graph into a junction tree. Although several heuristic-based graph transformations are suggested, a difficulty is to choose a set of heuristics so that clique size constraints are met. Also, there is no specific algorithm to construct the CT. A preferable method would be to make additions to an existing CT. Dynamic reconfiguration of CTs is explored (Darwiche, 1998), but it is specific to evidence and query based simplification. A more general approach using the Maximal Prime Subgraph Decomposition (MPD) of the PGM is discussed in Flores et al. (2002). In this method, the CT is converted into another graphical representation called the MPD join tree which is based on the moralized graph. When factors are added, the minimal subgraph of the moralized graph that needs re-triangulation is identified using the MPD tree. The identified subgraph is re-triangulated, and both the CT and MPD join trees are updated. In contrast, our method,

- Requires a lower effort for re-triangulation. This is because the minimal subgraph that is re-triangulated is not the modified moralized graph, but a portion of the modified chordal graph corresponding to the CT (which we have denoted as the elimination graph). Moreover, as opposed to Flores et al. (2002), the subgraph identified using our method need not always contain all variables present in the impacted cliques of the CT.

- Eliminates the memory and runtime requirements for maintaining additional representations like the moralized graph and the MPD join tree. Our method identifies the minimal subgraph to be re-triangulated directly from the CT, triangulates it and updates the CT. No other representation of the PGM is needed.

## 9 Discussion and Conclusions

We propose a technique for approximate inference of partition function that constructs a sequence of CTFs using a series of incremental build, infer and approximate steps. We prove the correctness of our incremental build and approximate algorithms.

IBIA gives better accuracies than several variational methods like LBP, region-graph based techniques like HAK, methods that simplify network like EDBP and WMB which is a mini-bucket based method. For the same clique size bound, accuracy obtained with IBIA is comparable or better than the neural network based methods DBE and NeuroBE in many cases, without having the disadvantage of requiring several hours of training. In most instances, the accuracy obtained with IBIA is comparable or better than recent sampling based techniques with much smaller runtimes. The runtimes are very competitive even though it is written in Python. Within a memory limit of 8 GB, IBIA was able to give PR estimates for 1705 of 1717 benchmarks. For a large percentage of these benchmarks, a solution was obtained within 20 minutes.

The main difficulty with predicting the performance of approximate inference algorithms is that besides being dependent on the graph structure of the PGM, it is also strongly dependent on the beliefs encoded by the PGM, which is what the algorithms are trying to estimate. In methods that use "loopy" cluster graphs, we generally expect better accuracies if the cluster(clique) sizes are larger since it accounts for a larger number of correlations between variables. However, inference methods that rely solely on the network structure to form clusters are not always useful. For example, both LBP and HAK are variants of iterative BP. While LBP uses minimum-sized clusters, HAK allows for the use of larger clusters to account for different cycle lengths. However, as seen in Table 6, the error obtained with HAK is larger than LBP in some cases. In contrast, approximations in IBIA are made based on both structure-based and belief-based information, resulting in lower errors than that obtained using the graph structure alone. However, as shown in Table 3, while belief-based metrics are useful in most cases, there are some cases where the error obtained using random selection is lower.

In IBIA, increasing the clique size bounds gives better accuracies in general. However, this also results in increased runtimes and memory utilization. IBIA constructs clique trees by incrementally adding factors to an existing CT. When the number of factors is large, repeated re-triangulations could increase the runtime. This is particularly seen in a few DBN instances where the number of factors is greater than 100,000 and the required runtime is around 100 minutes. Also, for benchmarks that have very large variable domain-sizes, the number of variables in cliques and sepsets in a CTF is small and approximation becomes difficult. This is seen in CSP benchmarks, where we were unable to solve 12 instances. Therefore, a good strategy is needed for the incremental build step that optimizes the runtime and results in reduced clique sizes. Approximation based on the $maxMI$ gives smaller error in most testcases. However, in a few Promedas and DBN testcases smaller errors were obtained with random selection, thus indicating a possibility for further exploration of heuristics.

A possible extension would be to combine IBIA and sampling based techniques to get accuracies that improve with time. The proposed IBIA framework can also be extended to handle other inference queries such as computation of the marginals, max-marginals and the most probable explanation. It also has implications in learning, since the tree-width limitations can be relaxed.

## Acknowledgements

We thank Prof. Rina Dechter and Bobak Pezeshki for providing reference values of the partition function for the large benchmarks for which exact solutions are not known.

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

## A  Proofs

**Propositions based on Algorithm 1:** Propositions 1 to 4 are based on our algorithm for the incremental addition of a group of factors $(\Phi_g)$ to an existing valid CTF using lines 3-15 of Algorithm 1. The modified CTF obtained after addition of factors is denoted as $CTF_m$. $SG_{min}$, $S_E$, $G_E$ and retained cliques are defined in Definitions $13-16$. All line numbers in these propositions refer to Algorithm 1.

**Proposition 1** $CTF_m$ contains only trees (possibly disjoint) i.e., no loops are introduced by the algorithm.

*Proof.* Algorithm 1 first identifies $SG_{min}$ for a set of factors that have overlapping minimal subgraphs in the input CTF (lines 4-6). $SG_{min}$ is either a single tree or a set of disjoint trees, each of which contains variables present in $\Phi_g$. The algorithm then constructs the elimination graph $G_E$ (refer Definition 16). When $SG_{min}$ has a single tree, $G_E$ is a connected graph because $S_E$ contains all the sepset variables in $SG_{min}$ and $G_E$ contains a fully connected component between $C \cap S_E$ for each clique $C$ in $SG_{min}$. When $SG_{min}$ has disjoint trees, $G_E$ gets connected when we add fully connected components corresponding to all factors in $\Phi_g$. Therefore, a single CT, $ST'$, is obtained on triangulating $G_E$ (line 21). Each retained clique either replaces or is re-connected to a single clique in $ST'$ (lines 25-29). Hence, the resulting structure $ST'$ continues to be a tree. Next, $SG_{min}$ is removed from the CTF (line 39). This results in disjoint trees, each containing a single clique in the adjacency list of $SG_{min}$. Each adjacent clique is re-connected to a single clique in the modified subtree $ST'$ (lines 40-42). Therefore, no loops are introduced and the modified CTF, $CTF_m$, continues to have one or more disjoint trees. □

**Proposition 2** $CTF_m$ contains only maximal cliques.

*Proof.* $ST'$ obtained after triangulating the elimination graph, $G_E$, has only maximal cliques by construction (line 21). The final $ST'$ is obtained after connecting retained cliques (lines 25-29), where there is an additional check for maximality. $CTF_m$ is obtained after replacing $SG_{min}$ with $ST'$. Cliques in the input CTF that are not in $SG_{min}$ contain at least one variable that is not present in $ST'$, thus remain maximal. □

**Proposition 3** All CTs in $CTF_m$ satisfy the running intersection property (RIP).

*Proof.* In Algorithm 1, $\mathcal{V}_{sg}$ is the set of variables in $SG_{min}$ and $\mathcal{V}_r$ is the set $\mathcal{V}_{sg} \setminus S_E$ (line 23). Consider the chordal graph corresponding to $SG_{min}$. This chordal graph has a perfect elimination order such that no fill-in edges are introduced on elimination. Even after the addition of edges between variables in the new factors, variables in $\mathcal{V}_r$ can be eliminated in this order without adding any fill-in edges. Therefore, cliques containing these variables are retained as is in the modified CTF, $CTF_m$. Elimination of variables in $S_E$ could potentially introduce fill-in edges as they are a part of chordless loops introduced by the addition of the new cliques. Therefore, only the subgraph of the modified chordal graph corresponding to the set of variables in $S_E$ needs re-triangulation. Construction of elimination graph $G_E$ using Definition 16 is equivalent to finding this subgraph. $ST'$ obtained after re-triangulation of $G_E$ is valid by construction and thus satisfies RIP. The modified subtree $ST'$ obtained after adding the retained cliques (lines 25-29) satisfies RIP because each retained clique $C$ is connected such that the sepsets contain all variables in the intersection $C \cap S_E$. This addition is always possible because $ST'$ is obtained from $G_E$ which contains a fully connected component between $C \cap S_E$ for all cliques $C$ in $SG_{min}$ (see Definition 16).

$CTF_m$ is obtained by removing $SG_{min}$ and reconnecting cliques adjacent to $SG_{min}$ in the input CTF ($Adj(SG_{min})$) to the cliques in $ST'$ (lines 36-44). Consider a clique $C_a$ in $Adj(SG_{min})$ that was connected via the set of sepset variables $S_a$. Since the input CTF satisfies RIP, $C_a \cap \mathcal{V}_{sg} = S_a$. In addition to $\mathcal{V}_{sg}$, the only variables in $ST'$ are variables in the new factors $\Phi_g$ which are not present in the input CTF. Therefore, $C_a \cap ST' = C_a \cap \mathcal{V}_{sg} = S_a$. Thus, RIP is satisfied since each adjacent clique $C_a$ is re-connected to a clique $C'$ in $ST'$ that contains the corresponding sepset $S_a$ (line 41). This connection is always possible because of the following reasons. If $C_a$ was connected to a retained clique, it can simply be re-connected via the same sepsets. Otherwise, if $C_a$ was connected to a clique $C$ in $SG_{min}$ that is not a retained clique, by construction every variable in $C$ must be present in $S_E$ i.e. $C \cap S_E = C$. Since $G_E$ contains a fully connected component corresponding to $C \cap S_E$ for all cliques in $SG_{min}$ (see Definition 16), $C$ is contained in at least one clique $C'$ in $ST'$ obtained after re-triangulation. Therefore, $C'$ also contains the sepset $S_a$ and can be connected to $C_a$. □

**Proposition 4** If the joint distribution captured by the input CTF with corresponding set of variables $X_{in}$ is $P(X_{in})$, then the joint distribution captured by $CTF_m$ is $P(X_{in}) \prod_{\phi \in \Phi_g} \phi$.

*Proof.* $CTF_m$ is obtained after replacing a subgraph of the existing CTF ($SG_{min}$) with a modified subtree $ST'$. We reassign the factors corresponding to each clique in $SG_{min}$ to cliques in $ST'$ containing their scope

(line 31). This is always possible because a) retained cliques in $SG_{min}$ are also present in $ST'$ and (b) cliques in $SG_{min}$ that are not retained cliques are contained in cliques in $ST'$ (as shown in the proof of Proposition 3). No change is made to factors assigned to the remaining cliques. The new factors in $\Phi_g$ are assigned to cliques in the modified CTF, $CTF_m$, containing the scope of these factors (line 32). This is possible because the elimination graph $G_E$ contains a fully connected components corresponding to all new factors (see Definition 16) and hence, these are contained in cliques obtained after re-triangulation. Therefore, if the product of factors in the input CTF is $P(X_{in})$, then the product of factors in $CTF_m$ is $P(X_{in}) \prod_{\phi \in \Phi} \phi$. $\qquad \square$

**Propositions based on Algorithm 2:** Propositions 5 to 8 are based on the proposed approximation algorithm (Algorithm 2) and Equations 4 and 5. We use $CTF_k$ to represent the input CTF to the algorithm and $CTF_{k,a}$ to represent the output of the algorithm. $CTF_k$ is a valid and calibrated CTF. IV denotes the set of interface variables (refer Definition 12) in $CTF_k$. All line numbers in these propositions refer to Algorithm 2.

**Proposition 5** All CTs in the approximate CTF, $CTF_{k,a}$, are valid CTs.

*Proof.* $CTF_{k,a}$ is initialized as $MSG[IV]$. $MSG[IV]$ is the minimal subgraph corresponding to the IVs (see Definition 11). Since all CTs in $CTF_k$ are valid, any connected subtree in $CTF_k$ is also valid. We now argue that all CTs in $CTF_{k,a}$ obtained after marginalization steps satisfy all properties of a valid CT.

- It contains only maximal cliques.
  Any non-maximal clique generated in the exact and local marginalization step is removed (lines 7 and 19).

- It contains disjoint trees.
  In the exact marginalization step, no loops are introduced since the neighbors of the collapsed and the non-maximal cliques are reconnected to $CTF_{k,a}$ so that connectivity of the CTs is preserved (lines 7 and 9). The local marginalization step only involves marginalization of variables from individual cliques and sepsets (line 18) and therefore does not alter the tree structure of the CTs in the CTF.

- It satisfies RIP.
  In the exact marginalization step, neighbors of all cliques that are collapsed are connected to the collapsed clique via corresponding sepsets and thus, RIP is satisfied. In the local marginalization step, variables are retained in a single connected subtree of $CTF_k$ (line 17-18). Also, in both steps, neighbors of non-maximal cliques which are removed are connected to the containing cliques with the same sepsets. Therefore, RIP is satisfied.

$\qquad \square$

**Proposition 6** All CTs in the approximate CTF, $CTF_{k,a}$, are calibrated.

*Proof.* In a calibrated CT, all adjacent cliques agree on the marginals over the sepset variables (see Equation 1). $CTF_{k,a}$ is obtained from $CTF_k$, which is calibrated. After exact marginalization, all the clique and sepset beliefs are preserved. Therefore, the resultant CTF is also calibrated. For the local marginalization step, let $v$ be the variable that is locally marginalized from adjacent cliques $C_i$ and $C_j$ with sepset $S_{i,j}$ in $CTF_k$. After local marginalization of variable $v$, we get the corresponding cliques $C_i'$ and $C_j'$ with sepset $S_{i,j}'$ in $CTF_{k,a}$, with the following beliefs (see Equation 5).

$$\beta(C_i') = \sum_{D_v} \beta(C_i), \quad \beta(C_j') = \sum_{D_v} \beta(C_j), \quad \mu(S_{i,j}') = \sum_{D_v} \mu(S_{i,j})$$

Here, $D_v$ is the domain of variable $v$. Since the result is invariant with respect to the order in which variables are summed out, we have

$$\sum_{Domain(C_i' \setminus S_{i,j}')} \beta(C_i') = \sum_{Domain(C_i' \setminus S_{i,j}')} \sum_{D_v} \beta(C_i) = \sum_{D_v} \sum_{Domain(C_i' \setminus S_{i,j}')} \beta(C_i) = \sum_{D_v} \mu(S_{i,j}) = \mu(S_{i,j}')$$

Similarly, $\sum_{Domain(C_j' \setminus S_{i,j}')} \beta(C_j') = \mu(S_{i,j}')$. Therefore, beliefs of the modified cliques $C_i'$ and $C_j'$ agree on the marginals of the sepset variables $S_{i,j}'$. Since this is true for every pair of adjacent cliques, all CTs in $CTF_{k,a}$ are calibrated. $\qquad\square$

**Proposition 7** The normalization constant of all CTs in the approximate CTF $CTF_{k,a}$ is the same as in the input CTF, $CTF_k$.

*Proof.* The approximation algorithm (Algorithm 2) has two steps, namely, exact marginalization and local marginalization. Exact marginalization involves collapsing cliques to find a joint belief (Equation 4) and then marginalizing a variable by summing over its states. Neither of these steps changes the normalization constant. Local marginalization involves marginalizing a variable from individual cliques and sepsets by summing over its states. Once again, it does not alter the normalization constant.

$\qquad\square$

**Proposition 8** If the clique beliefs are uniform, then the beliefs obtained after local marginalization are exact.

*Proof.* Let $C_1$ and $C_2$ be two adjacent cliques in $CTF_k$ with sepset $S_{1,2}$. After local marginalization of a variable $v$, we get the corresponding cliques $C_1'$ and $C_2'$ in $CTF_{k,a}$, with sepset $S_{1,2}'$. Let $b_1$, $b_2$ and $b_3$ represent the uniform beliefs in $C_1$, $C_2$ and $S_{1,2}$. If the domain of variable $v$ ($D_v$) has $k$ states, the beliefs of states in $C_1', C_2'$ and $S_{1,2}'$ are $kb_1, kb_2$ and $kb_3$.

The exact joint belief of $C_1$ and $C_2$ is

$$\beta(C_1 \cup C_2) = \frac{\beta(C_1)\beta(C_2)}{\mu(S_{1,2})}$$

Each state of $\beta(C_1 \cup C_2)$ has a constant belief $\frac{b_1 b_2}{b_3}$. If exact marginalization is carried out, the states of $\sum_{D_v} \beta(C_1 \cup C_2)$ have a constant belief $k\frac{b_1 b_2}{b_3}$. With local marginalization, the joint beliefs are

$$\beta(C_1' \cup C_2') = \frac{\beta(C_1')\beta(C_2')}{\mu(S_{1,2}')}$$

The corresponding constant beliefs are $\frac{(kb_1)(kb_2)}{kb_3} = k\frac{b_1 b_2}{b_3}$. $\qquad\square$

**Propositions based on inference of PR:** Proposition 9 and Theorem 2 relate to inference of partition function (PR).

**Proposition 9** Let the undirected graph associated with the PGM be connected and let $\{CTF_1, CTF_{1,a}, CTF_2, \ldots, CTF_{n-1,a}, CTF_n\}$ with the corresponding sets of variables $\{X_1, X_{1,a}, X_2 \ldots X_{n-1,a}, X_n\}$ denote the sequence of CTFs generated by Algorithms 1 and 2. Then, the normalization constant of the distribution encoded by $CTF_k$ ($Z_k$) is

$$Z_k = \begin{cases} \sum_{Domain(X_1)} \prod_{\phi \in \Phi_1} \phi & \text{for } k = 1 \\ \sum_{Domain(X_k)} \frac{\prod_{C' \in CTF_{k-1,a}} \beta(C')}{\prod_{SP' \in CTF_{k-1,a}} \mu(SP')} \prod_{\phi \in \Phi_k} \phi & \text{for } k > 1 \end{cases}$$

where, $\Phi_1, \ldots, \Phi_k$ are the subsets of initial factors added to $CTF_1, \ldots CTF_k$ respectively and

$$\sum_{Domain(X_{k-1,a})} \frac{\prod_{C' \in CTF_{k-1,a}} \beta(C')}{\prod_{SP' \in CTF_{k-1,a}} \mu(SP')} = \sum_{Domain(X_{k-1})} \frac{\prod_{C \in CTF_{k-1}} \beta(C)}{\prod_{SP \in CTF_{k-1}} \mu(SP)}$$

*Proof.* Since the infer step uses the standard belief propagation algorithm for exact inference to calibrate the CTs, the distribution encoded by $CTF_1$ is exactly $\prod_{\phi \in \Phi_1} \phi$. Therefore, the normalization constant of $CTF_1$ $(Z_1)$ is the following.

$$Z_1 = \sum_{Domain(X_1)} \prod_{\phi \in \Phi_1} \phi$$

The approximation algorithm (Algorithm 2) enforces the following two constraints (a) no CT in $CTF_{k-1}$ is disconnected (b) all interface variables are retained. These two constraints ensure that the number of CTs in $CTF_{k-1,a}$ is the same as that in $CTF_{k-1}$. This is because the first constraint ensures that the number of CTs in $CTF_{k-1,a}$ cannot be higher than that in $CTF_{k-1}$. The second constraint along with the assumption that input PGM is a connected graph, ensures that each CT in $CTF_{k-1}$ has at least one interface variable. If it is not so, it means that the PGM has a disconnected set of variables, which is a contradiction. Since all interface variables are retained, $CTF_{k-1,a}$ has a CT corresponding to each CT in $CTF_{k-1}$. Using Propositions 6 and 7, each CT in $CTF_{k-1,a}$ is calibrated and has the same normalization constant (NC) as the corresponding CT in $CTF_{k-1}$. The overall NC of the distribution encoded by $CTF_{k-1,a}$ is the product of NCs of all disjoint CTs. Therefore, the NC of $CTF_{k-1,a}$ is the same as that of $CTF_{k-1}$ i.e.

$$\sum_{Domain(X_{k-1,a})} \frac{\prod_{C' \in CTF_{k-1,a}} \beta(C')}{\prod_{SP' \in CTF_{k-1,a}} \mu(SP')} = \sum_{Domain(X_{k-1})} \frac{\prod_{C \in CTF_{k-1}} \beta(C)}{\prod_{SP \in CTF_{k-1}} \mu(SP)}$$

$CTF_k$ is built from $CTF_{k-1,a}$ by adding factors in $\Phi_k$ and then calibrated using belief propagation. Therefore, the NC of $CTF_k$ $(Z_k)$ can be written as follows.

$$Z_k = \sum_{Domain(X_k)} \frac{\prod_{C' \in CTF_{k-1,a}} \beta(C')}{\prod_{SP' \in CTF_{k-1,a}} \mu(SP')} \prod_{\phi \in \Phi_k} \phi$$

$\square$

**Theorem 2** Let the undirected graph corresponding to the PGM be connected and let the sequence $\{CTF_1, \cdots, CTF_n\}$ be the SCTF generated by IBIA. Then, the last CTF, $CTF_n$ contains a single CT, denoted as $CT_n$. IBIA returns the normalization constant of $CT_n$ $(Z_n)$ as the PR.

*Proof.* First, we show that Algorithm 1 (BuildCTF) with $mcs_p$ set to $\infty$ gives a single CT if the graph corresponding to the PGM is connected. Without loss of generality, assume an initial CTF, $CTF_0$ and a fixed order in which factors are added. $CTF_0$ contains disjoint CTs that have single cliques. Each factor that is added either modifies a CT or connects multiple CTs depending on the scope of the factor. Therefore, the number of CTs either decreases or remains the same as new factors are added. If there are disjoint CTs after all factors are added, it means that there is no factor in the PGM whose scope contains variables from each of the disjoint CTs. This means that the set of variables in these CTs are present in disjoint graphs of the PGM, which is not possible since the undirected graph corresponding to the PGM is connected by assumption. This is true for any initial set of cliques and any order in which factors are added.

If $mcs_p$ is set to a finite value, Algorithm 1 stops when this bound is reached. The CTs are then simplified and approximated by Algorithm 2. As argued in the proof of Proposition 9, $CTF_{k,a}$ and $CTF_k$ have the same number of CTs. Therefore, each CT in $CTF_{k,a}$ corresponds to a single CT in $CTF_k$ that has the same set of interface variables. Thus, when subsequent factors are added to the CTF in the same fixed order, the same CTs will get modified or connected. Since this is true of every approximate step, the final CTF will contain a single CT, denoted as $CT_n$. By Proposition 9, $CT_n$ is calibrated and has the normalization constant given by $Z_n$, which is returned by IBIA as the PR of the PGM. $\square$

