# OpenReview forum: "IBIA: An Incremental Build-Infer-Approximate Framework for Approximate Inference of Partition Function"
_TMLR — Accepted by TMLR_

### Review · Reviewer_M91U · 2023-06-05

**Summary Of Contributions:**

A new method (IBIA) for approximate computation of the partition
function (for a graphical model) is given. The method is explained in
sufficient detail using well-chosen examples. Much related work is
cited and briefly described. The value of the method rests
overwhelmingly on whether it provides reasonable approximations in a
reasonable time (as compared to existing methods), so the deciding
factor is empirical performance. Empirical comparisons against a
number of reasonable competing approaches are done, and sure enough,
IBIA does well enough to be considered a SOTA method.

**Audience:**

Yes

**Claims And Evidence:**

Yes

**Requested Changes:**

The main requested changes I have were mentioned with "strengths and weaknesses". Here are some additional small points:

p1: #P completeness does not necessitate approximation in all cases,
only in general

p4: SCTF : perhaps remind reader at this point what "SCTF" means.

p13: typo "CT Fk+1 is constructed by adding new factors are added"

Table 6: "Entries are marked as '-' where all instances could not
solved". This might confuse the reader since they might (reasonably)
interpret this as: "no instances were solved". I would rephrase to
"where not all instances could be solved" or "at least one instance
could not be solved".


**Strengths And Weaknesses:**

Although I have a generally positive view of this work - an effective
method for addressing an important problem is presented - there are
some improvments that could be made.

It is only when we get to the "Related Work" section that I got some
idea of how the authors came up with this method - it's a development
in the "incremental construction of CTs" line of work. I think it
would be better if this came earlier and if the reader were given some
idea of why/how incremental construction of CTs work. At present, the
paper is addressed at those already working in this area (which will
indeed by its main audience); I think a bit more explanation would
enlarge the prospective audience considerably. For example, just
adding a sentence explicitly stating how a calibrated clique tree
allows recovery of the partition function would be useful.

This is a "present a method and empirically test it" paper. There is
little theory explaining the impact of the various choices in IBIA;
the method has the look of something that was derived by trial and
error. A there useful theoretical results available in other work on
incremental construction of CTs? If so, it would be worth flagging up
here.

It's a pity the authors chose to implement in Python. As they note
this makes it harder to do useful empirical comparisons to methods
implemented in e.g. C/C++. In actual applications all one cares about
is performance (accuracy/speed) and less about whether this comes from
clever method, native code execution or good coding. I know this is
not a concern of the current paper, but it may be that gains are
availaible by sticking with IBIA but working on good implementation.

---

> ### Author Response · Authors · 2023-06-07
> **Response to review by reviewer M91U (part 1)**
>
> We would like to thank you for your comments and suggestions. Our response to each question/suggestion is shown below.
>
> > It is only when we get to the "Related Work" section that I got some idea of how the authors came up with this method - it's a development in the "incremental construction of CTs" line of work. I think it would be better if this came earlier and if the reader were given some idea of why/how incremental construction of CTs work.
>
> Actually, incremental construction of CTs is only one part of our inference technique.
> It is used so that we effectively divide the PGM into multiple sections, each of which has a bounded clique size.
>
> In our paper, the section on related work is a little fragmented. We had a detailed comparison with related work for each part of our algorithm (incremental construction and approximation) without giving a unified picture of where our algorithm fits. The proposed technique falls under the category of "methods that use multiple CTs for inference'' rather than incremental construction alone.  To make this clear, we could add the following subsection in Section 3 that explains the motivation of our approach, before giving the overview of the method. This material was presented in related work but the motivation was lost.
>
> *Section 3.2 Motivation*
>
> Since the complexity of inference is exponential in the maximum clique size, the key to making the problem tractable is to bound the clique size. Typically, bounding clique sizes leads to loopy graphs and convergence issues. An alternative is to divide the PGM into multiple sections such that each section results in a CT with smaller clique sizes making it amenable to non-iterative belief propagation. None of the existing approaches in this category have been used for inference of partition function. Following are the two main challenges that need to be addressed: (a) How do we divide the PGM such that the maximum clique size of each CT is less than a user-specified bound? and (b) How do we exchange beliefs between the CTs, so that the overall partition function can be inferred?
>
> Previous attempts at dividing the PGM include the exact inference method, multiply sectioned BN (MSBN) (Xiang et al., 1993; Xiang \& Lesser, 2003), and the approximate inference method in Bhanja \& Ranganathan (2004). In MSBNs, the network structure is divided into sections. The CT for each section is built using co-operative triangulation and there is no guarantee that the maximum clique size will be within a specified bound. Bhanja \& Ranganthan (2004) use an iterative method for partitioning that requires repeated conversion of different candidate sections to corresponding CTs, which is compute-intensive.
>
> Since each section contains a subset of factors in the PGM, we need a way of exchanging beliefs between CTs so that the overall partition function can be computed.
> In Bhanja \& Ranganthan (2004), the information is transmitted from one CT to the next using an approximate Chow-Liu tree, containing variables present in both CTs.
> A sequence of CTs is also obtained for the 2T-BN model of dynamic BNs in the method proposed in Murphy (2002). Beliefs are transferred from one CT to the next using joint beliefs over subsets of variables present in both CTs (Boyen \& Koller, 1998). However, both methods do not allow for the estimation of the overall partition function. A more detailed discussion of why it is not possible is presented in Section 8.
>
> These two challenges are addressed in this paper.
>
> > This is a ``present a method and empirically test it" paper. There is little theory explaining the impact of the various choices in IBIA; the method has the look of something that was derived by trial and error. A there useful theoretical results available in other work on incremental construction of CTs? If so, it would be worth flagging up here.
>
> We think that if we add material detailing the motivation of our approach, it will help dispel the notion that it is something derived by trial and error.
>
> Regarding theoretical results: Our overall algorithm has three parts and for each part, we provide guarantees that are detailed in the propositions. These guarantees allow for the inference of the partition function. For example,  we guarantee that our incremental build algorithm always results in a valid clique tree and our approximation algorithm preserves the partition function.
>
> Since there are no other methods involving multiple CTs that can be used for inference of partition function, we do not have anything to compare with or any other theory that can be added.

---

> ### Author Response · Authors · 2023-06-09
> **Response to review by reviewer M91U (part 2)**
>
> Contd.
>
> > It's a pity the authors chose to implement in Python. As they note this makes it harder to do useful empirical comparisons to methods implemented in e.g. C/C++. In actual applications all one cares about is performance (accuracy/speed) and less about whether this comes from clever method, native code execution or good coding. I know this is not a concern of the current paper, but it may be that gains are available by sticking with IBIA but working on good implementation.
>
> Regarding implementation in Python, we want to point out that in spite of it being implemented in Python, it is faster than many other inference algorithms implemented in C/C++. As seen in the results section, the number of instances solved by IBIA ($mcs_p=20$) in 20s is much larger than double-loop GBP (HAK) and sample search with the same clique size bound. We used Numpy libraries for some of the time-critical portions, so it is fast. But yes, we should be able to get further gains if implemented in C/C++.
>
>  > At present, the paper is addressed at those already working in this area (which will indeed be its main audience); I think a bit more explanation would enlarge the prospective audience considerably. For example, just adding a sentence explicitly stating how a calibrated clique tree allows recovery of the partition function would be useful.
>
> This property of a calibrated clique tree was stated in the background section. But, we can re-iterate it in the overview section to make it more comprehensible.

---

### Review · Reviewer_uA1W · 2023-06-20

**Summary Of Contributions:**

This work proposes an sequential approach to approximating the partition function of a graphical model. This involves constructing a sequence of clique tree forests (SCFTs) by incorporating factors while satisfying a maximal clique size constraint (a user parameter). The paper also describes an approximation algorithm that takes in a clique tree forest, and outputs another with a reduced clique size, also a user specified parameter. The partition function of the last clique tree is the estimate of the partition function of the graphical model. The authors prove some properties of their algorithms, discuss complexity and apply it to a number of benchmark datasets.

**Audience:**

Yes

**Claims And Evidence:**

Yes

**Requested Changes:**

While the authors do a good job presenting the details of their algorithm, the paper would be of interest to a larger audience if the authors better motivated how the ideas presented here fit into the existing literature (e.g. are there other CTF methods?), what exactly are the novel ideas here, when and why they might be expected to do well or outperform existing methods. Please see my comment on weaknesses above.

**Strengths And Weaknesses:**

__Strengths:__
The problem the authors consider is an important one, and the proposed algorithm seems quite novel to me. The paper itself is very clearly organized, and despite the complexity of the algorithm, it quite understandable: I especially appreciate the running example provided. The experimental evaluation also seems reasonably thorough.

__Weaknesses:__
My biggest concern with the paper is that while the authors do a good job describing their algorithm, I (as a nonexpert in this field) have very little intuition for why this should work well, what limitations of existing methods this addresses, how the parameters of the algorithm affect accuracy, and even why we expect this to be more accurate than other methods. The paper would benefit from a much more thorough intuition of where the ideas presented come from, rather than just describing them.

Similarly, while the experiments are quite thorough, it might help to study more qualitatively a synthetic experiment where we can look at a particular graphical model(s), and see how its properties interact with the properties of the algorithm. Right now, the benchmarking experiments do not supplement my understanding of the algorithm, they just make the case (convincingly) that the algorithm is useful.

Proposition 9 and Theorem 10 say that the normalization constants of the CTF(s) are the "estimate" of the true partition function. I am confused by this language. Do the authors actually mean that they are exactly the true partition function? More broadly, it would help to better clarify (at least roughly) when the algorithms output (the estimate) is exact, and when it is approximate.

---

> ### Author Response · Authors · 2023-06-27
> **Response to review by reviewer uA1W (part2)**
>
> Contd.
>
> **Experiments to study the interaction between properties of the graphical model and properties of IBIA:**
> While we can come up with a synthetic graphical model to illustrate a testcase where IBIA works well or badly, the performance of IBIA could be very different for another PGM with the same graph structure but different initial beliefs. This is also valid for other approximate inference methods since besides being dependent on the graph structure of the PGM, the performance of inference algorithms is also strongly dependent on the beliefs encoded by the PGM, which is what the algorithms are trying to estimate.
> In methods that use ``loopy" cluster graphs, we generally expect better accuracies if the cluster(clique) sizes are larger since it accounts for a larger number of correlations between variables. However, inference methods that rely solely on the network structure to form clusters are not always useful. For example, both LBP and HAK are variants of iterative BP. While LBP uses minimum-sized clusters, HAK allows for the use of larger clusters to account for different cycle lengths. However, as seen in Table 6, the error obtained with HAK is larger than LBP in some cases.
>
> In contrast, approximations in IBIA are made based on both structure-based and belief-based information, resulting in lower errors than that obtained using the graph structure alone. However, as shown in Table 3, while belief-based metrics are useful in most cases, there are some cases where the error obtained using random selection is lower.
>
> We can add this discussion to Section 9 (Discussion and Conclusions).
>
> **Conditions under which IBIA gives the exact solution:** When SCTF has a single CTF, the PR obtained is exact.
> If the SCTF has multiple CTFs, it is still possible to get the exact PR if the approximate step uses only exact marginalization. But this
> is rare and in most cases, local marginalization is required, and the PR obtained is approximate.
>
>
> We can add this to Section 6 after complexity analysis.

---

> ### Author Response · Authors · 2023-06-27
> **Response to review by reviewer uA1W (part1)**
>
> We would like to thank you for your comments and suggestions. Our response to each question/suggestion is shown below.
>
> **Intuition of where the ideas presented come from, limitations of existing methods that use multiple CTFs:**
>  Looking at the paper again, we feel that our section on related work is a little fragmented and the motivation of our approach is lost. To make it clear, we can add a subsection in Section 3 (Overview of the IBIA paradigm) that describes the motivation of the approach before giving an overview of IBIA. To summarize,
>
>  None of the existing approaches that use multiple CTFs have been used for the inference of partition function (PR).
> Following are the two main issues that need to be addressed by methods in this category.
>
> (a) How do we divide the PGM into sections such that the maximum clique size of each CTF is less than a user-specified bound?
>
> Previous attempts at dividing the PGM either cannot guarantee clique size bounds (Xiang et al., 1993; Xiang \& Lesser, 2003; Murphy, 2002) or use an iterative approach that requires repeated re-triangulation of different candidate sections to find corresponding CTFs, which is compute-intensive.
>
> (b) How do we exchange beliefs between the CTFs so that the overall partition function can be inferred?
>
> The information is transferred from one CTF to the next using approximate Chow-Liu trees in Bhanja \& Ranganathan (2004) and using joint beliefs over clusters of variables in Murphy (2002); Boyen \& Koller (1998). Both approaches are used for transferring normalized beliefs and cannot be used for the estimation of PR.
>
> **How IBIA addresses the limitations of existing methods that use multiple CTFs:**
> To build CTFs with bounded clique sizes, IBIA uses an incremental strategy that requires re-triangulation of only a portion of the CTF in each step. This reduces the time complexity of the approach. In IBIA, the beliefs are exchanged from one CTF to the next using the approximate CTF. Our approximation strategy preserves the PR (Proposition 7) which allows for inference of the overall PR from the last CTF in the sequence.
>
> We can re-write the section on related work to make this more clear.
>
> **How parameters affect accuracy:** IBIA uses two parameters $mcs_p$ and $mcs_{im}$. Increasing $mcs_p$ increases the clique size bound for each CTF. This in turn could potentially reduce the number of CTFs and the number of approximate steps, thus improving accuracy. Also, the metrics used for guiding the approximate step are computed using beliefs corresponding to the partial set of factors added up to the current CTF. Therefore, the accuracy of the metrics could improve when a larger set of factors is added, resulting in better estimates. In the paper, Table 4 shows a comparison of results obtained with different values of $mcs_p$.
>
> $mcs_{im}$ sets the clique size bounds on the approximate CTF. The impact of this parameter is discussed in `Choice of parameters' (Section 7, Page 14).
>
> **Why will it work better than existing approaches:** Most of the existing variational methods for PR estimation perform iterative BP on loopy cluster graphs that are constructed based on the graph structure corresponding to the PGMs. In contrast, approximations in IBIA are based on both belief-based and structure-based information which results in better accuracies. Moreover, since IBIA uses clique trees instead of cluster graphs, the belief propagation step is non-iterative, which makes it fast. We have mentioned these points in our discussion on the limitations of existing work and contributions of this paper (Page 2).
>
> As mentioned in Section 8 (Related work), none of the existing approaches that use multiple CTFs can be used for the approximate inference of the partition function.

---

### Review · Reviewer_Tkfo · 2023-07-01

**Summary Of Contributions:**

The paper proposes new method(s) to approximate the partition function by recursively constructing clique trees.

**Audience:**

Yes

**Broader Impact Concerns:**

None.

**Claims And Evidence:**

Yes

**Requested Changes:**

I have added them in the weakness section.

**Strengths And Weaknesses:**

Strengths:

The use of a particular example to explain the algorithm is very helpful.

The empirical results are promising, and exhaustive.

Weaknesses:

The algorithms are themselves quite simple. This is not necessarily a weakness in itself, especially considering good empirical performance, and the fact that it has not been done before, but the presentation despite the algorithms being simple is severely lacking. The background section, and the algorithm description needs a lot of work. I would not have been able to understand the jargon without reading other papers and the example in section 4.1.1. The theoretical "contribution" is too hand wavy.


Fig 2: why is mcs_{im} 3? CTK_2 still has cliques of size 4, no ?

Prop 5 is redundant, the “validity” as stated follows trivially.

Prop 8: isn’t the “uniformity” beliefs a very strong assumption ? why is this not assumption not practically limiting ?

Prop 9 and theorem 2 are too hand wavy, there is no concrete approximation. The use of phrases like “estimate of” and “depends on” adds nothing of value to the actual understanding of how good or bad theoretically the proposed algorithm could be. Please remove or make it more concrete.

Background and clarity in writing:
The background section needs some more work. If the authors are drilling into basics (which it seems they are given the attempt to define even what a PGM is), the background could use more clarity.

In background: please mention \alpha is a set of indices.

The construction of graphs and cliques does not become clear till Sec 3.3. Please clarify in the background itself if each node of the graph is a factor or that each node can contain more than just a factor.  The definition of PGM merely says PGM is a collection of factors, the correspondence between factors and nodes is not obvious. This is confusing because of the following clash in the notation: in the RIP property of the CT, it says X \in C_i (does this mean X is contained in some node of C_i ? or that X itself is a node in the click C_i ?), and then says X is present in every node in the unique path between C_i and C_j. The latter statement seems to imply X \in C_i overloads the symbol “\in” to mean that X is in some node in C_i, which probably means each node is a factor.

Eq (1) is ill-formed. It should be \sum_{x \in C_i \S_{I,j}} \beta (x).

The belief function \beta(.) springs out of nowhere for anyone who has no background in PGMs. Please write a couple of sentences writing what it is, and how it relates to \phi. Otherwise the statement about clique beliefs in calibrated CT having the same Z equal to partition function makes no sense.

If the authors want to assume background for the reader, please just skip the background entirely. If the authors want to not assume any background, please flush it out well.

The validity of CTF (definition 8) seems redundant. If CTF is a set of CTs, can you give an example of an invalid CTF ? Isnt a CTF by definition valid?

Clique size: again what does it mean variable v \in Clique ? Does it mean variable in a node in the clique ? what is “state” of a variable (is not mentioned anywhere else)? Are all variables discrete ?

I have no idea what definition 12 means, is there a typo? Intersection of variables in CTK_k and set of variables present in factors that have not been added to CTK_i where i can also be k ?

---

> ### Author Response · Authors · 2023-07-07
> **Response to review by reviewer Tkfo (part 1)**
>
> We would like to thank you for your comments and suggestions. Our response to each question/suggestion is shown below.
> We have also uploaded a revised manuscript that addresses these concerns.
>
> > Fig 2: why is $mcs_{im}$ 3? $CTK_2$ still has cliques of size 4, no?
>
> As discussed in the overview of IBIA (Section 3.3 of the revised manuscript), $mcs_{im}$ is the maximum clique size for the approximate CTF, $CTF_{k,a}$. The maximum clique size for each CTF in the SCTF, $CTF_k$,  is $mcs_p$ which is set to 4 in the example. Therefore, the maximum clique size in both $CTF_1$ and $CTF_2$ is 4.
>
> > Prop 5 is redundant, the “validity” as stated follows trivially.
>
> Proposition 5 basically says that Algorithm 2 will always result in a valid CTF that satisfies RIP. The proof looks at each step of the algorithm and shows that the output CTF is always valid. We feel that it is important to show this because if we do not have a valid CTF, we cannot proceed further.
>
> In this paper, we have a series of algorithms and we want to show that each algorithm gives some guarantees. These guarantees allow for the inference of PR.
>
> > Prop 8: isn’t the “uniformity” beliefs a very strong assumption ? why is this  assumption not practically limiting ?
>
> We first want to emphasize that uniformity of beliefs is not an assumption of our method. What we are saying is that if clique beliefs happen to be uniform, then our algorithm will give the exact PR.
> As pointed out by the reviewer, this is rarely the case and IBIA gives approximate estimates in most cases.
>
> > Prop 9 and theorem 2 are too hand wavy, there is no concrete approximation. The use of phrases like “estimate of” and “depends on” adds nothing of value to the actual understanding of how good or bad theoretically the proposed algorithm could be. Please remove or make it more concrete.
>
> Each CTF in the SCTF, $CTF_k$, is constructed by adding a partial set of factors in the PGM. But, the normalization constant of beliefs in $CTF_k$ accounts for factors added to it and the previous CTFs, $\{CTF_1,...,CTF_k\}$. *This allows for the inference of PR from the last CTF in the sequence.* Therefore, these propositions are non-trivial and are needed for the estimation of PR.
> However, we can replace sentences in the proof of Proposition 9 with corresponding equations to make it more clear (as shown in Appendix A of the revised manuscript, Page 30).
>
> The phrase ``estimate of" is used to highlight the fact that the value obtained using IBIA need not be exact.  The equations in the revised proof (refer Page 30) should make it clear why it is an estimate.
> While we agree there are no theoretical bounds on the error of estimates obtained with IBIA, this is also true for most of the other variational inference methods like LBP, GBP, HAK, IJGP, DBE, etc. One method that does give an upper bound is WMB, but the bounds are very loose. As seen from Table 7, the errors obtained using this method are very large.
>
> > I have no idea what definition 12 means, is there a typo? Intersection of variables in $CTK_k$ and set of variables present in factors that have not been added to $CTK_i$ where i can also be k ?
>
> There is no typo and the definition is correct. But, we can re-write it as follows, if it makes it more clear.
>
> Let $\Phi=\{\Phi_1,...,\Phi_n\}$ where $\Phi_k$ denotes the set of factors added during the construction of $CTF_k$. A variable in $CTF_k$ is an interface variable if it is present in the scope of any factor in the set $\{\Phi_{k+1}, ...,\Phi_n\}$.

---

> ### Author Response · Authors · 2023-07-07
> **Response to review by reviewer Tkfo (part 2)**
>
> Contd.
>
> > Background and clarity in writing: The background section needs some more work. If the authors are drilling into basics (which it seems they are given the attempt to define even what a PGM is), the background could use more clarity.
> If the authors want to assume background for the reader, please just skip the background entirely. If the authors want to not assume any background, please flush it out well.
>
> We want to keep the background and make it as comprehensible as possible. Thank you for your comments and suggestions. We can add more details to make it more clear.
>
> > The construction of graphs and cliques does not become clear till Sec 3.3. Please clarify in the background itself if each node of the graph is a factor or that each node can contain more than just a factor. The definition of PGM merely says PGM is a collection of factors, the correspondence between factors and nodes is not obvious.
>      In background: please mention $\alpha$ is a set of indices.
>
> We can modify the definition of PGMs as follows. $Z=\sum_{Domain(\mathcal{X})} \prod_{\alpha}\phi_{\alpha}$
>
> Let  $\mathcal{X} = \{ X_1, X_2, \cdots X_n\}$ be a set of random variables with associated domains $D = \{D_{X_1},D_{X_2}, \cdots D_{X_n}\}$.
>     The probabilistic graphical model (PGM) over $\mathcal{X}$ consists of a set of factors, $\Phi$.  Each factor $\phi_{\alpha}(\mathcal{X_\alpha}) \in \Phi$  is defined over a subset of variables $Scope(\phi_\alpha)=\mathcal{X_\alpha}$ where $\alpha$ denotes the index to the set of factors.
> The domain $D_\alpha$ of $\mathcal{X_\alpha}$  is the Cartesian product of the domains of variables in $\mathcal{X_\alpha}$ and the factor $\phi_\alpha$ is a map $\phi_\alpha: D_\alpha \rightarrow R \geq 0$.
> The joint probability distribution captured by the model is $P(\mathcal{X})=\frac{1}{Z} \prod_\alpha \phi_\alpha$  where the normalizing constant, $Z=\sum_{Domain(\mathcal{X})} \prod_\alpha \phi_\alpha$ is the partition function (PR).
>
> Each node of the undirected graph corresponding to the PGM is associated with a random variable. Variables $X_i$ and $X_j$ are connected via an edge in this graph if there is at least one factor in the PGM ($\Phi$) whose scope contains both variables.
>
>
> > This is confusing because of the following clash in the notation: in the RIP property of the CT, it says $X \in C_i$ (does this mean X is contained in some node of $C_i$ ? or that X itself is a node in the click $C_i$ ?), and then says X is present in every node in the unique path between $C_i$ and $C_j$. The latter statement seems to imply $X \in C_i$ overloads the symbol “$\in$” to mean that X is in some node in $C_i$, which probably means each node is a factor.
>
> The undirected graph associated with the PGM is converted to a chordal graph ($\mathcal{H}$). The cliques are fully connected components in $\mathcal{H}$. Since nodes in the graph associated with the PGM are random variables, each clique $C_i$ contains a set of variables. Cliques form the nodes of the clique tree model.
>
> We hope that the revised definition of the PGM makes the subsequent definitions clearer.
>
> > Eq (1) is ill-formed. It should be $\sum_{x \in C_i \setminus S_{I,j}} \beta (x)$.
>
> Equation (1) is correct. This is the way it is often written in textbooks, see for example Equation 10.7 (Page 358) in Koller \& Friedman (2009). However, to avoid confusion, we can change it as follows.
>
> $\sum\limits_{Domain(C_i\setminus S_{ij})} \beta(C_i) = \sum\limits_{Domain(C_j\setminus S_{ij})} \beta(C_j) = \mu(S_{ij})$
>
> > The belief function $\beta(.)$ springs out of nowhere for anyone who has no background in PGMs. Please write a couple of sentences writing what it is, and how it relates to $\phi$. Otherwise, the statement about clique beliefs in calibrated CT having the same Z equal to partition function makes no sense.
>
> We can add the following definition of clique beliefs before defining a calibrated CT.
>
> After two-way message passing, each clique in the CT has an associated joint belief $\beta(C_i) =\sum_{Domain(\mathcal{X}\setminus C_i)} ~\prod_\alpha \phi_{\alpha}$.
> and each sepset has an associated joint belief $\mu(S_{ij}) =\sum_{Domain(\mathcal{X}\setminus S_{ij})} \prod_\alpha \phi_{\alpha}$.
>
> > The validity of CTF (definition 8) seems redundant. If CTF is a set of CTs, can you give an example of an invalid CTF ? Isnt a CTF by definition valid?
>
> A valid CTF is a set of valid CTs, each of which has to satisfy the conditions elaborated in definition 5. Definition 8 just states this.
>
> > Clique size: again what does it mean variable v $\in$ Clique ? Does it mean variable in a node in the clique ? what is “state” of a variable (is not mentioned anywhere else)? Are all variables discrete ?
>
> Does the revised definition of the PGM make this clearer?
>
> Yes, all variables are discrete. States in the domain of variable mean the values that a variable can take.

---

### Decision · Action_Editors · 2023-08-12

**Recommendation:** Accept with minor revision

**Comment:**

I suggest that Proposition 9 and Theorem 2's statements be made formal, for example the word "estimate of" doesn't immediately imply a formal meaning. I want to suggest that the statements be made mathematically formal, and the current texts could be tagged as informal statements as the current text is helpful in conveying the intuition but not formal enough to verify the proof precisely. Also, the proofs need to be revised; for example, the proof of Theorem 2 says: "we ensure that CTs
remain connected while approximation" but it's not clear how this implies the previous line. I would suggest that authors expand on the proof.

**Audience:**

Yes, given the fundamental nature of the problem.

**Claims And Evidence:**

The paper focuses on a fundamental problem in probabilistic inference: the computation of partition function. The methodologies argued in the paper focuses on reducing a graphical model into sequence of clique tree forests with bounded clique sizes, for which computation is efficient. The reviewers appreciated the empirical contributions but shared their concern regarding the need for rigor in theoretical analysis.

---

> ### Author Response · Authors · 2023-09-19
>
> We would like to thank the reviewers and the action editors for their time and effort in reviewing the paper. The valuable feedback provided has helped us improve the quality of the paper. We have incorporated the suggested changes and uploaded the camera ready version of the paper.
>
> Thank you,
> Authors

---

> > ### Comment · Action_Editors · 2023-09-19
> > **Camera Ready version**
> >
> > The camera-ready version has incorporated the required revisions.
> >
> > Congratulations to the authors for a really nice piece of work.